# Discrete Diffusion with Physical Mass Constraints for *De Novo* Peptide Sequencing

**Zeyu An**[1]  **Wanyu Lin**[1]

## Abstract

*De novo* peptide sequencing is a pivotal technique that directly reconstructs amino acid sequences from tandem mass spectrometry (MS/MS) data; it enables the identification of novel proteins and variants absent from reference databases. Previous methods are typically based on autoregressive (AR) decoding or one-shot generation. The AR-based methods conflict with the bidirectional and globally constrained nature of MS/MS evidence and inevitably accumulate errors, while one-shot generation does not explicitly enforce physical constraints, failing to produce chemically valid and reliable peptides in a single pass. Accurate sequencing necessitates reasoning over the entire peptide simultaneously, enabling iterative self-correction under global constraints. To this end, we introduce **PhysNovo**, a novel paradigm that harnesses discrete diffusion to enable simultaneous global reasoning and iterative refinement. Specifically, PhysNovo reformulates sequencing as a **phys**ically mass-constrained reasoning process by embedding a knapsack-based feasibility kernel to enforce exact precursor mass consistency. By conditioning the diffusion process on global spectral context, PhysNovo supports abductive reasoning where bidirectional evidence is exploited to iteratively resolve local inconsistencies and ensure physically valid predictions. PhysNovo achieves state-of-the-art performance, exceeding baselines by over 2% in precision, with larger gains on out-of-distribution data. The source code is publicly available at https://github.com/WanyuGroup/ICML2026_PhysNovo.

[1]Department of Computing, The Hong Kong Polytechnic University, Hong Kong SAR, China. Correspondence to: Wanyu Lin <wan-yu.lin@polyu.edu.hk>.

*Proceedings of the 43$^{rd}$ International Conference on Machine Learning*, Seoul, South Korea. PMLR 306, 2026. Copyright 2026 by the author(s).

## 1. Introduction

Peptide sequencing via tandem mass spectrometry constitutes a cornerstone of modern proteomics, underpinning protein characterization, biomarker discovery, and therapeutic development (Aebersold & Mann, 2003; Lee et al., 2007; Macklin et al., 2020; Strauss et al., 2024; Wenk et al., 2024). Traditionally, this task is addressed through database search methods, which match observed spectra against a fixed catalog of known reference peptides. While widely adopted, this paradigm suffers from a fundamental limitation: it can only recover sequences that are explicitly present in the predefined database (Noor et al., 2021). Consequently, it is inherently incapable of identifying novel peptides, such as unexpected mutations or rare variants, that are absent from existing libraries (Bandeira et al., 2008). To bridge this gap, *de novo* peptide sequencing has emerged as a critical computational approach. By reconstructing amino acid sequences directly from MS/MS data independent of reference databases, this method enables the discovery of novel proteins beyond the boundaries of traditional libraries (Hettich et al., 2013; Tran et al., 2017).

Prevailing *de novo* methods (Tran et al., 2017; Qiao et al., 2021; Eloff et al., 2023; Yilmaz et al., 2024; Yang et al., 2024; Chen et al., 2025) formulate sequencing as a next-token prediction problem, effectively casting it as a sequence-to-sequence translation task. While these AR paradigms enable fast inference, they suffer from a inductive bias mismatch with the physical generation process of mass spectrometry (Qiao et al., 2021). Physically, MS/MS fragmentation provides global, bidirectional evidence: spectral fragment ion (MS2) masses corresponding to both sequence prefixes and suffixes appear jointly constrain the peptide structure (Zhou et al., 2024). In contrast, AR decoding enforces a left-to-right causal order. This reduces a globally constrained inference problem to a sequence of greedy, irreversible local decisions, where early errors propagate downstream regardless of future contradictory evidence. Even emerging non-autoregressive (NAR) paradigm (Zhang et al., 2025), while mitigating sequential error propagation, relies on one-shot generation that struggles to guarantee reliable predictions in a single pass. Crucially, these existing paradigms fail to intrinsically internalize physical laws;

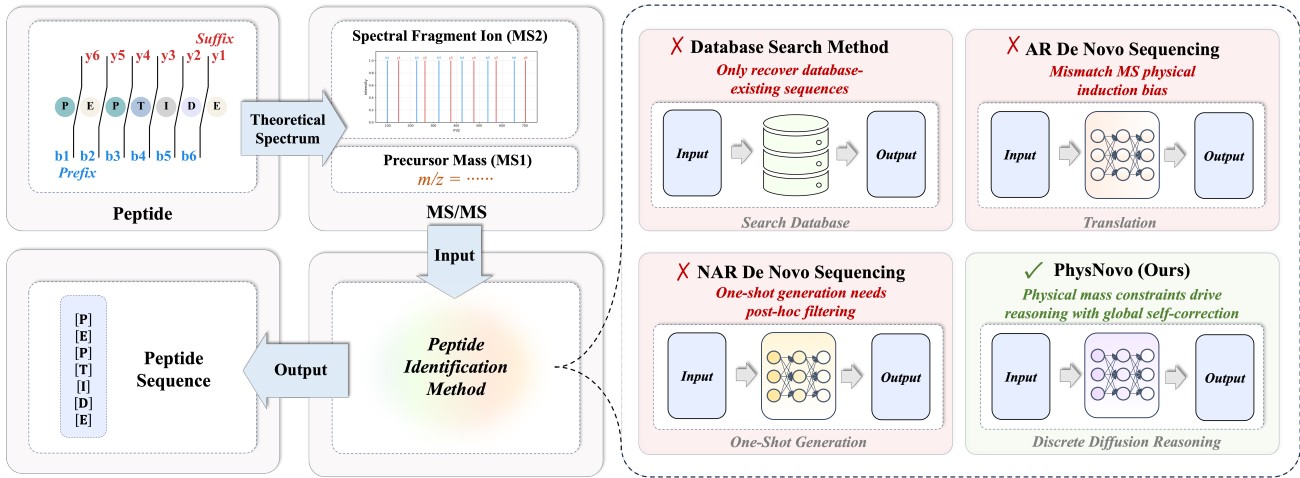

*Figure 1.* Conceptual comparison of peptide identification paradigms.

specifically, the precursor mass (MS1, the measured total mass of the peptide) imposes a strict equality constraint on the sum of residue masses. Both AR and NAR models frequently generate chemically invalid sequences that violate this mass conservation, requiring inefficient post-hoc filtering (Yilmaz et al., 2024) (see Appendix B for a detailed discussion). These limitations suggest that standard linguistic translation formulations represent an oversimplification of *de novo* sequencing (Figure 1). We argue that this process necessitates a paradigm shift toward a global reasoning task.

To realize this, we introduce **PhysNovo**, a paradigm that reframes *de novo* peptide sequencing as a **phys**ically mass-constrained discrete diffusion process. Leveraging the nature of this paradigm, PhysNovo generates the peptide sequence through iterative refinement. This allows each denoising step to act as a global consistency check against both MS1 information and MS2 evidence. Our core methodological innovation lies in bridging generative modeling with combinatorial optimization: we embed a knapsack-based feasibility kernel directly into the diffusion transition distribution. This ensures that every intermediate hypothesis sampled during inference strictly satisfies the MS1 constraint, effectively reducing the search space to the manifold of chemically valid sequences. Simultaneously, the diffusion process is conditioned on the full MS2 spectrum, allowing the model to perform global error correction-revising locally incorrect residue assignments that conflict with the overall spectral context. This iterative verification process guarantees that PhysNovo produces physically valid and reliable sequences, overcoming the inherent unreliability of AR and one-shot generation.

Our main contributions are summarized as follows:

- **Generative reasoning with physical constraints.** We propose a discrete diffusion paradigm that shifts peptide

sequencing from a greedy translation to an iterative denoising process. This formulation naturally aligns with the bidirectional nature of MS/MS data, enabling joint reasoning over fragment evidence from both sequence prefixes and suffixes.

- **Hard mass-constrained discrete diffusion.** We introduce a novel diffusion mechanism that enforces exact precursor mass conservation at every inference step. We solve the issue of generating chemically invalid peptides without relying on heuristic post-processing.

- **Global error correction capability.** By conditioning the full diffusion trajectory on spectral evidence, PhysNovo enables abductive reasoning—detecting and correcting local inconsistencies through global spectral alignment. This mechanism effectively mitigates the sequential error accumulation inherent in baselines, allowing for self-correction to ensure prediction reliability.

- **State-of-The-Art performance.** Extensive experiments on standard benchmarks demonstrate that PhysNovo achieves state-of-the-art performance. It significantly outperforms baselines in amino acid, peptide, and post-translational modification (PTM) -level accuracy, while exhibiting superior reliability and generalization to unseen species and low-resolution spectra.

## 2. Related Work

### 2.1. De Novo Peptide Sequencing

Early deep learning approaches for *de novo* sequencing, such as DeepNovo (Tran et al., 2017) and SMSNet (Karunratanakul et al., 2019), treated the task as sequence translation using CNNs and LSTMs, while PointNovo (Qiao et al., 2021) later introduced set-based architectures. Recently, the field has converged on Transformer-based AR models (Yilmaz et al., 2024; Eloff et al., 2023), with subsequent works

enhancing performance through auxiliary mechanisms (Xia et al., 2024; Yang et al., 2024), latent imputation (Du et al., 2025), or retrieval frameworks (Chen et al., 2025). Most recently, $\pi$-PrimeNovo (Zhang et al., 2025) introduced NAR decoding to accelerate inference and mitigate sequential error propagation. However, existing paradigms remain limited: AR models suffer from unidirectional bias, while NAR approaches typically operate as one-shot generators lacking iterative refinement. Crucially, both fail to intrinsically internalize physical laws during generation, treating precursor mass (MS1) merely as a post-hoc filter rather than an active constraint, often yielding chemically invalid predictions.

### 2.2. Physically Constrained Generation by Diffusion

Generative diffusion models have achieved remarkable success in scientific domains, particularly for continuous structures like proteins (Watson et al., 2023) and small molecules (Xu et al., 2022; Bohde et al., 2025). For discrete data, Discrete Denoising Diffusion Probabilistic Models (D3PM) (Austin et al., 2021) and Masked Diffusion (Sahoo et al., 2024) have emerged as powerful alternatives to AR models for text and protein sequence generation (Wang et al., 2024; 2025; Hayes et al., 2025). Nevertheless, imposing strict physical constraints in discrete diffusion remains an open challenge. While approaches like DiffMS (Bohde et al., 2025) successfully enforce chemical formula constraints for small molecule graphs using a transition-based method, they are tailored for unordered graph generation and do not generalize to the sequential, ordered nature of peptides required for proteomics. Existing protein language models focus primarily on folding stability or functional properties but lack mechanisms to enforce the exact mass conservation essential for mass spectrometry. PhysNovo bridges this gap by establishing a generative paradigm capable of enforcing exact precursor mass conservation throughout the global iterative denoising of peptide sequences.

## 3. Methodology

PhysNovo reframes *de novo* peptide sequencing as a physically constrained discrete diffusion process (Figure 2), constructing the peptide via iterative refinement of a global hypothesis. The framework proceeds in three stages: (1) **Structure-Aware Pretraining**. We pretrain on enzyme-digested Swiss-Prot (UniProt Consortium, 2018) sequences without MS/MS data to learn biochemical priors. (2) **MS/MS-Constrained Fine-Tuning**. The model is fine-tuned on benchmark training splits, integrating soft MS2 alignment and MS1 conditioning. (3) **Inference as Iterative Hypothesis Refinement**. Starting from a fully masked sequence, PhysNovo recovers the peptide identity through an iterative, physically constrained reverse diffusion process, strictly enforcing hard MS1 mass constraints and global

MS2 spectral consistency. Section 3.1 outlines preliminaries, followed by details in Sections 3.2–3.4.

### 3.1. Preliminaries

**Problem Formulation.** *De novo* peptide sequencing aims to reconstruct the amino acid sequence of a peptide solely from its MS/MS data, without reliance on reference databases. Let $\mathcal{V}$ denote the vocabulary of amino acids (residues), including the 20 canonical amino acids and pertinent PTMs. A peptide sequence is represented as a discrete vector $\mathbf{y} = (y_1, y_2, \ldots, y_L) \in \mathcal{V}^L$, where $L$ denotes the peptide length, which is a prior unknown.

**Spectral Data and Fragmentation (MS2).** The input to the sequencing model consists of the observed fragment spectrum, commonly referred to as the MS2 data. Formally, this spectrum is defined as a set of peaks $\mathbf{x} = \{(m_i, I_i)\}_{i=1}^{N}$, where each tuple represents the mass-to-charge ratio $(m/z)$ $m_i \in \mathbb{R}^+$ and the intensity $I_i \in \mathbb{R}^+$ of a detected ion fragment. Physically, these peaks result from the stochastic fragmentation of the peptide backbone, typically generating prefix ions ($b$-ions) and suffix ions ($y$-ions) that retain the N- or C-terminal distinct parts of the sequence, respectively.

**Precursor Mass and Physical Constraints (MS1).** Associated with the MS2 spectrum is the precursor information $\mathbf{c} = (m_{\text{prec}}, z)$ obtained from MS1. This tuple comprises the observed precursor $m/z$ value $m_{\text{prec}}$ and the charge state $z$. A fundamental physical law governing this process is the principle of mass conservation. To rigorously define this constraint for the generative process, we derive the target residue mass $M_{\text{res}}$ by removing the charge carriers (protons) and the terminal water molecule from the precursor:

$$M_{\text{res}} = (m_{\text{prec}} - m_{H^+}) \cdot z - w(\text{H}_2\text{O}), \qquad (1)$$

where $m_{H^+}$ is the proton mass and $w(\cdot) : \mathcal{V} \to \mathbb{R}$ maps a residue to its monoisotopic mass. Consequently, a valid peptide hypothesis $\mathbf{y}$ must satisfy the global constraint that the sum of its constituent residues equals this target mass within a specific tolerance $\epsilon$:

$$\left| \sum_{l=1}^{L} w(y_l) - M_{\text{res}} \right| \leq \epsilon. \qquad (2)$$

Unlike formulations that decompose the generation into local conditional probabilities $P(y_l \mid \mathbf{y}_{<l}, \mathbf{x})$, we frame *de novo* sequencing as learning the joint posterior distribution $p(\mathbf{y} \mid \mathbf{x}, \mathbf{c})$. Our goal is to sample $\mathbf{y}$ from this distribution to maximize spectral consistency while strictly satisfying the mass constraint in Eq. (2).

### 3.2. Structure-Aware Pretraining via Discrete Diffusion

To enable the model to reason about peptide structures before associating them with MS/MS data, we first perform

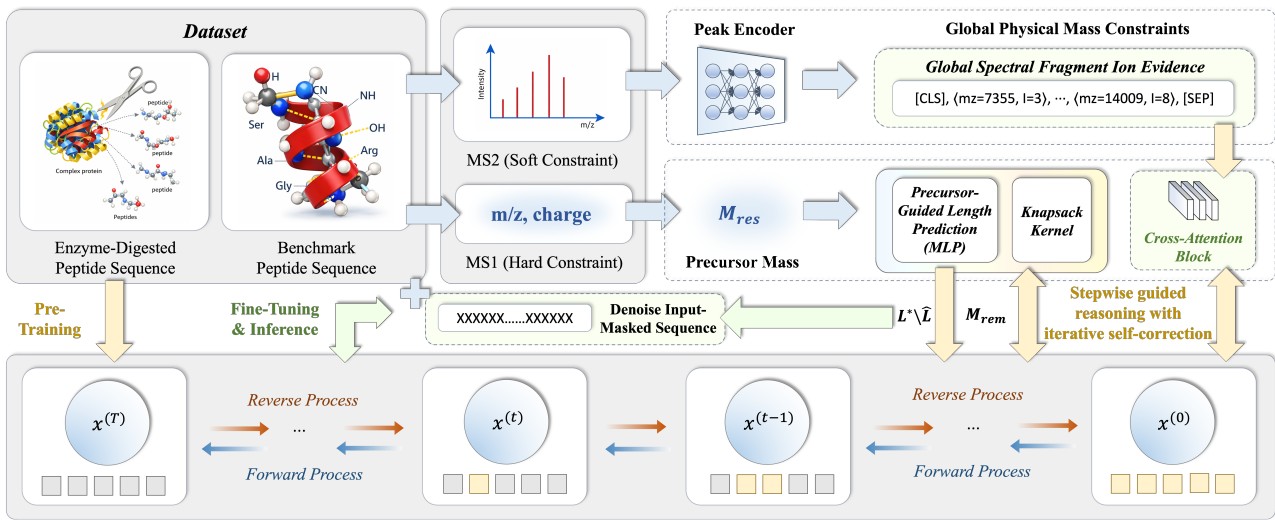

*Figure 2.* The schematic overview of the PhysNovo framework. The framework integrates a discrete diffusion backbone with a Knapsack-based Hard Mass Constraint and Spectral-Guided Error Correction, enabling iterative reasoning for de novo peptide sequencing.

unconditional pretraining to learn the peptide prior $p(\mathbf{y})$. Following DPLM (Wang et al., 2024), we formulate this as a discrete diffusion process on the amino acid vocabulary $\mathcal{V}$. The forward process $q(\mathbf{y}_t \mid \mathbf{y}_{t-1})$ progressively corrupts $\mathbf{y}_0$ by masking residues according to a variance schedule $\beta_t$ until reaching a fully masked state at $t = T$. The generative task learns the reverse process $p_\theta(\mathbf{y}_{t-1} \mid \mathbf{y}_t)$ parameterized by a network $\Phi_\theta$, optimizing the negative log-likelihood of masked tokens:

$$\mathcal{L}_{\mathrm{pre}}(\theta) = \mathbb{E}_{t, \mathbf{y}_0 \sim \mathcal{D}_{\mathrm{pre}}} \left[ -\sum_{l \in \mathcal{M}_t} \log p_\theta(y_{0,l} \mid \mathbf{y}_t, t) \right], \quad (3)$$

where $\mathcal{M}_t$ denotes the set of indices masked at step $t$, and $\mathcal{D}_{\mathrm{pre}}$ represents the peptide corpus described below. This objective forces the model to reconstruct peptide fragments based on bidirectional context, effectively learning the grammar of amino acid compositions.

**Network Architecture.** Our backbone $\Phi_\theta$ employs a 12-layer bidirectional Transformer encoder, following the architectural design of ESM-2 (Lin et al., 2023) (adapted to our vocabulary). To accommodate the variable lengths inherent in peptide sequences, we define a maximum length $L_{\mathrm{max}}$ and pad all sequences to this dimension using a special [PAD] token. We incorporate a binary attention mask into the self-attention mechanism, ensuring that the model attends strictly to valid residues while excluding padding tokens from both the receptive field and the loss calculation. Unlike AR and NAR models that cast sequencing as a translation task, our bidirectional architecture captures global dependencies between N-terminal and C-terminal residues simultaneously. This global receptive field is crucial for the subsequent stages, as physical fragmentation

evidence is distributed across the entire sequence. Note that at this stage, the cross-attention modules intended for MS/MS conditioning are disabled, and the model relies exclusively on self-attention to internalize the compositional syntax of peptide sequences.

**Biochemical Priors via *In Silico* Digestion.** To align with mass spectrometry physics, we construct the pretraining dataset $\mathcal{D}_{\mathrm{pre}}$ by performing *in silico* digestion on the Swiss-Prot (UniProt Consortium, 2018) database. We simulate cleavage rules of common enzymes (Trypsin and Lys-C) to generate biologically realistic peptide fragments, rather than using raw protein sequences. This prevents distribution shift and instills a strong prior for chemically plausible peptides, significantly reducing the search space for subsequent inference (dataset details in Sec. 4.1).

### 3.3. MS/MS-Constrained Fine-Tuning

We now fine-tune the model to learn the posterior $p_\theta(\mathbf{y} \mid \mathbf{x}, \mathbf{c})$, transforming the prior $p(\mathbf{y})$ into a spectral reasoner. We optimize a joint objective comprising three components.

#### 3.3.1. PRECURSOR-GUIDED LENGTH PREDICTION

Since discrete diffusion requires a fixed-dimensional noise tensor, we must estimate the peptide length $L$ from the precursor information $\mathbf{c} = (m_{\mathrm{prec}}, z)$ prior to inference. We formulate this as a classification problem over a range of plausible lengths $[L_{\mathrm{min}}, L_{\mathrm{max}}]$. A lightweight prediction head $f_\phi$, parameterized by a multi-layer perceptron (MLP), maps $\mathbf{c}$ to a categorical distribution:

$$P_\phi(L \mid \mathbf{c}) = \mathrm{Softmax}\left(\mathrm{MLP}(\mathbf{c})\right). \quad (4)$$

During fine-tuning, we minimize the cross-entropy loss for the ground truth length $L^*$:

$$\mathcal{L}_{\text{len}}(\phi) = -\log P_\phi(L = L^* \mid \mathbf{c}). \quad (5)$$

This ensures that at inference time, we can initialize the diffusion state $\mathbf{y}_T$ with the most probable length $\hat{L} = \text{argmax}_L P_\phi(L \mid \mathbf{c})$.

### 3.3.2. CONSTRAINT I: SOFT SPECTRAL ALIGNMENT

To incorporate the fragmentation evidence, we encode the MS2 spectrum $\mathbf{x} = \{(m_i, I_i)\}_{i=1}^N$ into a latent sequence of peak embeddings $\mathbf{H}_\mathbf{x} \in \mathbb{R}^{N \times d}$. We employ the Sinusoidal PeakEncoder (Yu & Li, 2025) to map continuous spectral data into the model's vector space.

Specifically, the PeakEncoder handles mass and intensity as follows: First, the $m/z$ value $m_i$ is mapped to a $d$-dimensional structural embedding $\mathbf{e}_{m_i}$ using multi-scale sinusoidal functions, allowing the model to perceive mass differences at varying resolutions:

$$\mathbf{e}_{m_i}[j] = \begin{cases} \sin\left(m_i \cdot 10000^{-2k/d}\right) & \text{if } j = 2k, \\ \cos\left(m_i \cdot 10000^{-2k/d}\right) & \text{if } j = 2k+1. \end{cases} \quad (6)$$

Second, the intensity $I_i$ is projected and added to the mass embedding to form the final peak representation $\mathbf{h}_i$:

$$\mathbf{h}_i = \mathbf{e}_{m_i} + I_i \cdot \mathbf{w}_I, \quad (7)$$

where $\mathbf{w}_I \in \mathbb{R}^d$ is a learnable parameter vector.

To bridge the structural prior with spectral evidence, these embeddings $\mathbf{H}_\mathbf{x}$ are injected into the backbone via Cross-Attention layers. We compute context-aware representations by attending peptide hidden states ($\mathbf{H}_\mathbf{y}^{(l)}$) to spectral peaks:

$$\text{CrossAttn}(\mathbf{H}_\mathbf{y}^{(l)}, \mathbf{H}_\mathbf{x}) = \text{softmax}\left(\frac{(\mathbf{H}_\mathbf{y}^{(l)}\mathbf{W}_Q)(\mathbf{H}_\mathbf{x}\mathbf{W}_K)^\top}{\sqrt{d}}\right)(\mathbf{H}_\mathbf{x}\mathbf{W}_V) \quad (8)$$

This mechanism imposes a soft constraint: high attention weights are assigned to peaks that physically support the presence of specific amino acids, effectively steering the denoising trajectory toward spectrally consistent sequences.

### 3.3.3. CONSTRAINT II: MASS CONDITIONING

To enforce the global mass budget, we condition the diffusion process on the target residue mass $M_{\text{res}}$ (Eq. 1). We add a precursor embedding $\mathbf{e}_\mathbf{c} = \text{MLP}(M_{\text{res}} \oplus z)$ to the timestep embedding, implicitly guiding the model toward mass-compliant sequences.

The final objective combines the conditional diffusion loss and length prediction:

$$\mathcal{L}_{\text{total}} = \mathbb{E}_{t,\mathbf{x},\mathbf{c}}\left[-\sum_{l \in \mathcal{M}_t} \log p_\theta(y_{0,l} \mid \mathbf{y}_t, \mathbf{x}, \mathbf{c}, t)\right] + \lambda\mathcal{L}_{\text{len}}. \quad (9)$$

where $\lambda$ balances the two tasks. By optimizing Eq. (9), PhysNovo learns to generate sequences that are not only structurally valid (from pretraining) but also spectrally consistent and compliant with the precursor mass.

### 3.4. Inference as Iterative Hypothesis Refinement

In the final stage, we perform *de novo* sequencing by bridging generative modeling with combinatorial optimization. PhysNovo initializes a global hypothesis and refines it through physically constrained discrete diffusion. We implement this via: a Knapsack-Based Feasibility Kernel for MS1 mass conservation, and a Global Error Correction strategy for MS2 spectral alignment.

**Hard Mass Constraint (MS1) via Knapsack Kernel.** To strictly enforce the mass conservation law (Eq. (2)) derived from MS1 data, we embed a knapsack-style combinatorial check directly into the diffusion transition. At each denoising step $t$, when sampling a residue for position $j$, we verify whether the choice allows the sequence to remain on the valid mass manifold defined by the target residue mass $M_{\text{res}}$.

Let $\mathbf{y}_{\text{fixed}}$ be the set of currently determined residues and $R$ be the number of remaining masks. For a candidate amino acid $a$, the remaining mass budget is calculated as:

$$M_{\text{rem}}(a) = M_{\text{res}} - \left(\sum_{y \in \mathbf{y}_{\text{fixed}}} w(y) + w(a)\right). \quad (10)$$

We formulate this as a combinatorial feasibility problem to determine if the remaining mass $M_{\text{rem}}(a)$ is reachable by any valid combination of $R - 1$ residues. We approximate the solution by checking the bounded reachability:

$$(R-1) \cdot w_{\min} - \epsilon \leq M_{\text{rem}}(a) \leq (R-1) \cdot w_{\max} + \epsilon, \quad (11)$$

where $w_{\min}$ and $w_{\max}$ are the minimum and maximum residue masses in $\mathcal{V}$, and $\epsilon$ is the mass tolerance. If this condition fails, token $a$ is explicitly masked by setting its logit to $-\infty$. This kernel ensures that every intermediate hypothesis effectively prunes the search space, guiding the generation strictly along chemically valid pathways.

**Global Error Correction via MS2 Spectral Conditioning.** While the Knapsack kernel guarantees strict MS1 compliance, it does not ensure alignment with MS2 fragment ions. Furthermore, the stochastic sampling process is susceptible to local generation errors. To rectify these, we employ a spectral-guided remasking strategy. Throughout the inference process, we monitor the hypothesis to identify residues where the model's predicted probability falls below a confidence threshold $\tau$. Specifically, for any position $j$ where the prediction confidence $p_\theta(y_j \mid \mathbf{x}) < \tau$, the residue is considered uncertain and forcibly reset to [MASK]. This mechanism enables abductive reasoning where the model leverages the bidirectional context to iteratively revise its decisions by deliberately erasing suspicious parts of the se-

quence. Unlike one-shot decoding that accepts the initial output, this iterative verification ensures high prediction reliability by actively resolving local ambiguities.

# 4. Experiments

## 4.1. Experimental Setup

**Pretraining Data**. To instill the model with biochemical structural priors, we construct an unlabeled peptide corpus derived from the Swiss-Prot dataset (UniProt Consortium, 2018). Following standard proteomics protocols, we perform *in silico* enzymatic digestion on the protein sequences using Trypsin and Lys-C cleavage rules. We apply filters to retain biologically relevant peptides with lengths $L \in [7, 40]$. After removing duplicates, this process yields a corpus of 15,535,514 unique peptide sequences. This dataset is used strictly for unconditional generative pretraining to learn the amino acid grammar and has no spectral overlap with the downstream test sets (details in Appendix D.1).

**Benchmark Datasets**. To ensure a comprehensive evaluation across varying spectral resolutions, biological origins, and chemical modifications, we evaluate PhysNovo on three established benchmarks (Zhou et al., 2024): Nine-species (Tran et al., 2017), Seven-species (Tran et al., 2017), and HC-PT (Eloff et al., 2023). The Nine-species and Seven-species datasets represent high- and low-resolution spectral conditions, respectively, while the HC-PT dataset provides a rigorous testbed for human-origin peptides and isoforms. Consistent with standard protocols (Zhou et al., 2024), we employ a cross-species evaluation strategy for the multi-species datasets, using *Saccharomyces cerevisiae* (yeast) exclusively for testing to assess generalization. Detailed descriptions of dataset statistics, PTM configurations, and split protocols are provided in Appendix D.2.

**Evaluation Metrics**. We employ a comprehensive evaluation protocol spanning three granularities: peptide, amino acid, and PTM. (1) **Peptide-Level Precision & AUC**: These serve as the primary indicators of practical utility. Precision measures the fraction of perfectly reconstructed sequences, while the Area Under the Precision-Recall Curve (AUC) evaluates the reliability of the model's confidence estimation and ranking capability. (2) **Amino Acid (AA) Precision & Recall**: These provide a fine-grained assessment of residue-level reconstruction accuracy based on mass and prefix/suffix alignment. (3) **PTM Precision & Recall**: These metrics quantify the model's robustness in identifying and localizing post-translational modifications. For all metrics, In all cases, Leucine (L) and Isoleucine (I) are treated as equivalent due to their isomeric nature. Detailed definitions and calculation protocols are provided in Appendix D.3.

**Implementation Overview.** PhysNovo employs a 12-layer Bidirectional Transformer backbone. The training pro-

ceeds in two stages: structure-aware pretraining followed by MS/MS-constrained fine-tuning. For inference, we use $T = 100$ discrete denoising steps with hard mass constraints ($\epsilon = 0.02$ Da) and global spectral correction ($\tau = 0.7$). Comprehensive details regarding model architecture, inputs and vocabulary, training protocols, inference configuration and mass definitions are provided in Appendix D.4.

## 4.2. Comparison with State-of-The-Arts

We present a comprehensive performance comparison against nine representative baselines, ranging from traditional algorithms to state-of-the-art deep learning models: PEAKS (Ma et al., 2003), DeepNovo (Tran et al., 2017), PointNovo (Qiao et al., 2021), InstaNovo (Eloff et al., 2023), CasaNovo (Yilmaz et al., 2024), AdaNovo (Xia et al., 2024), $\pi$-HelixNovo (Yang et al., 2024), LIPNovo (Du et al., 2025), $\pi$-PrimeNovo (Zhang et al., 2025), and ReNovo (Chen et al., 2025). As summarized in Table 1, the results demonstrate that PhysNovo establishes a new state-of-the-art across all three benchmarks, effectively overcoming the limitations of the current AR and NAR methods through physically mass constrained global discrete diffusion.

**Peptide-Level Performance.** PhysNovo consistently outperforms leading baselines in peptide sequencing accuracy. Notably, on the challenging Seven-species dataset, it achieves 0.346 precision, exceeding ReNovo (0.278) and LIPNovo (0.327) by absolute margins of 6.8% and 1.9%, respectively. It maintains this lead on high-resolution benchmarks (Nine-species, HC-PT). Furthermore, PhysNovo improves peptide AUC on Seven-species by 6.2% over ReNovo, demonstrating robust full-sequence recovery and ranking capability across diverse spectral resolutions.

**Amino Acid-Level Performance.** At the fine-grained level, PhysNovo also exhibits robust improvements. On Nine-species, PhysNovo improves precision over LIPNovo by 1.8% and recall by 1.7%. Crucially, it outperforms the recent NAR baseline $\pi$-PrimeNovo (0.790) by 2.5%, validating the superiority of iterative reasoning over one-shot decoding. On HC-PT, PhysNovo attains the highest recall (0.652); while ReNovo shows marginally higher precision, PhysNovo's balanced recall and peptide-level dominance suggest more effective overall sequencing.

**PTM-Level Performance.** Table 2 details the performance on identifying modified residues, which is often confounded by subtle mass shifts. PhysNovo outperforms all baselines across every metric and dataset. On Seven-species, it surpasses LIPNovo by 5.0% in precision and 5.1% in recall. Similar gains are observed on high-resolution data (e.g., +3.1% precision on HC-PT vs. LIPNovo). These consistent improvements confirm superior sensitivity in localizing subtle mass shifts, essential for identifying structural variations.

*Table 1.* Comparison with state-of-the-art methods on Nine-species, Seven-species, and HC-PT datasets in amino acid-level and peptide-level performance. The best results are marked in **bold**, and the second best are underlined.

| Method | Amino Acid-Level Performance | | | | | | Peptide-Level Performance | | | | | |
| | Nine-species | | Seven-species | | HC-PT | | Nine-species | | Seven-species | | HC-PT | |
| | Prec. | Recall | Prec. | Recall | Prec. | Recall | Prec. | AUC | Prec. | AUC | Prec. | AUC |
|---|---|---|---|---|---|---|---|---|---|---|---|---|
| PEAKS (Ma et al., 2003) | 0.748 | - | - | - | - | - | 0.428 | - | - | - | - | - |
| DeepNovo (Tran et al., 2017) | 0.696 | 0.638 | 0.492 | 0.433 | 0.531 | 0.534 | 0.428 | 0.376 | 0.204 | 0.136 | 0.313 | 0.255 |
| PointNovo (Qiao et al., 2021) | 0.740 | 0.671 | 0.196 | 0.169 | 0.623 | 0.622 | 0.480 | 0.436 | 0.022 | 0.007 | 0.419 | 0.373 |
| InstaNovo (Eloff et al., 2023) | 0.420 | 0.395 | 0.192 | 0.176 | 0.289 | 0.285 | 0.164 | 0.123 | 0.031 | 0.009 | 0.057 | 0.034 |
| CasaNovo (Yilmaz et al., 2024) | 0.697 | 0.696 | 0.322 | 0.327 | 0.442 | 0.453 | 0.481 | 0.439 | 0.119 | 0.084 | 0.211 | 0.177 |
| AdaNovo (Xia et al., 2024) | 0.698 | 0.709 | 0.379 | 0.385 | 0.442 | 0.451 | 0.505 | 0.469 | 0.174 | 0.135 | 0.212 | 0.178 |
| $\pi$-HelixNovo (Yang et al., 2024) | 0.765 | 0.758 | 0.481 | 0.472 | 0.588 | 0.582 | 0.517 | 0.453 | 0.234 | 0.173 | 0.356 | 0.318 |
| LIPNovo (Du et al., 2025) | 0.797 | 0.797 | 0.557 | 0.560 | 0.637 | 0.643 | 0.582 | 0.547 | 0.327 | 0.281 | 0.458 | 0.427 |
| $\pi$-PrimeNovo (Zhang et al., 2025) | 0.790 | - | - | - | - | - | - | - | - | - | - | - |
| ReNovo (Chen et al., 2025) | 0.770 | 0.769 | 0.512 | 0.514 | **0.651** | 0.648 | 0.568 | 0.528 | 0.278 | 0.228 | 0.467 | 0.436 |
| **PhysNovo (Ours)** | **0.815** | **0.814** | **0.568** | **0.569** | 0.649 | **0.652** | **0.588** | **0.553** | **0.346** | **0.290** | **0.473** | **0.444** |

*Table 2.* Comparison with state-of-the-art methods on Nine-species, Seven-species, and HC-PT datasets in PTM-level performance.

| Method | Nine-species | | Seven-species | | HC-PT | |
| | Prec. | Recall | Prec. | Recall | Prec. | Recall |
|---|---|---|---|---|---|---|
| DeepNovo | 0.576 | 0.529 | 0.391 | 0.373 | 0.626 | 0.615 |
| PointNovo | 0.629 | 0.546 | 0.117 | 0.094 | 0.676 | 0.740 |
| InstaNovo | 0.443 | 0.294 | 0.126 | 0.115 | 0.350 | 0.261 |
| CasaNovo | 0.706 | 0.566 | 0.360 | 0.251 | 0.501 | 0.460 |
| AdaNovo | 0.652 | 0.570 | 0.448 | 0.321 | 0.552 | 0.482 |
| $\pi$-HelixNovo | 0.680 | 0.598 | 0.473 | 0.366 | 0.568 | 0.667 |
| LIPNovo | 0.765 | 0.656 | 0.604 | 0.498 | 0.732 | 0.745 |
| **PhysNovo (Ours)** | **0.793** | **0.686** | **0.654** | **0.549** | **0.763** | **0.770** |

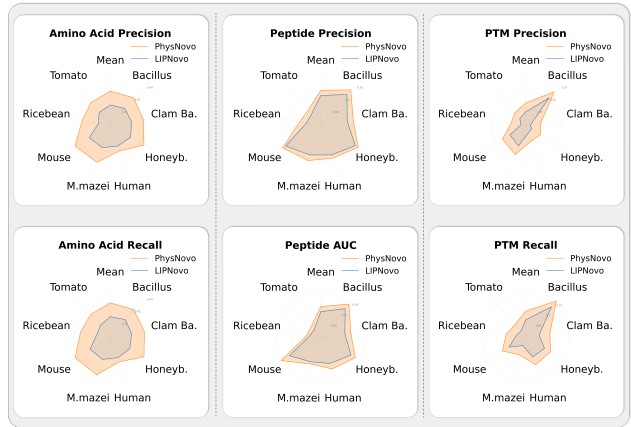

*Figure 3.* Visualization of generalization capability via leave-one-out cross-validation on the Nine-species dataset. We compare **PhysNovo (orange)** against **LIPNovo (blue)** across six metrics.

**Leave-One-Out Cross-Validation.** To assess generalization to unseen biological domains, we perform leave-one-out cross-validation on the Nine-species dataset. As shown in Table 3 and Figure 3, PhysNovo consistently outperforms LIPNovo across all species. On average, it yields gains of 1.6% in AA precision and 2.6% in PTM precision. This uniformity confirms robust generalization to unseen distributions, ensuring reliable performance for open-world proteomic discovery.

**Parameters vs. Model Performance.** As analyzed in Table 4, while LIPNovo relies on parameter scaling (68.4M) to improve over CasaNovo (47.4M), PhysNovo achieves State-of-the-Art results with 63.1M parameters (5.3M fewer than LIPNovo). By enforcing hard mass constraints to prune the invalid search space, PhysNovo reduces the model's burden to memorize chemical rules, achieving higher accuracy via structural efficiency rather than brute-force expansion.

**Inference Efficiency.** Beyond parameter efficiency, practical deployment in high-throughput proteomics hinges on inference speed. Table 5 evaluates the inference throughput on a single RTX 4090 GPU. Compared to the autoregressive baseline (CasaNovo), PhysNovo achieves a substantial speedup alongside significant accuracy gains (0.815

vs. 0.697). While our iterative refinement process introduces latency compared to single-step NAR methods like $\pi$-PrimeNovo, it delivers superior precision. Crucially, the discrete diffusion framework naturally allows for a flexible speed–accuracy trade-off: reducing the sampling steps to $T = 50$ yields near-NAR throughput (99 vs. 105 spectra/s) while preserving highly competitive accuracy (0.801 vs. 0.790). This demonstrates that PhysNovo is not only structurally efficient but also computationally practical for large-scale analysis.

### 4.3. Reliability and Confidence Calibration

Beyond accuracy metrics, practical proteomics demands model reliability. This is a critical weakness in standard AR and one-shot NAR models, which lack internal verification. Figure 4 presents the recall-coverage curves (also known as precision-coverage curves) for PhysNovo across three benchmarks. Crucially, PhysNovo exhibits a robust high-shoulder characteristic: the curves maintain > 95% amino

*Table 3.* Leave-one-out cross-validation performance on the Nine-species dataset compared against the state-of-the-art Method.

| Species | Method | Amino Acid | | Peptide | | PTM | |
|---|---|---|---|---|---|---|---|
| | | Prec. | Recall | Prec. | AUC | Prec. | Recall |
| Bacillus | LIPNovo | 0.806 | 0.807 | 0.607 | 0.581 | 0.855 | 0.739 |
| | **PhysNovo** | 0.822 | 0.822 | 0.614 | 0.589 | 0.878 | 0.765 |
| Clam Ba. | LIPNovo | 0.805 | 0.805 | 0.591 | 0.563 | 0.764 | 0.647 |
| | **PhysNovo** | 0.819 | 0.820 | 0.598 | 0.571 | 0.794 | 0.685 |
| Honeyb. | LIPNovo | 0.807 | 0.806 | 0.606 | 0.577 | 0.769 | 0.675 |
| | **PhysNovo** | 0.825 | 0.825 | 0.610 | 0.584 | 0.800 | 0.699 |
| Human | LIPNovo | 0.805 | 0.805 | 0.596 | 0.567 | 0.772 | 0.672 |
| | **PhysNovo** | 0.811 | 0.810 | 0.600 | 0.575 | 0.797 | 0.701 |
| M.mazei | LIPNovo | 0.807 | 0.807 | 0.596 | 0.565 | 0.812 | 0.630 |
| | **PhysNovo** | 0.825 | 0.826 | 0.603 | 0.568 | 0.839 | 0.664 |
| Mouse | LIPNovo | 0.808 | 0.807 | 0.607 | 0.579 | 0.803 | 0.667 |
| | **PhysNovo** | 0.827 | 0.827 | 0.611 | 0.592 | 0.828 | 0.694 |
| Ricebean | LIPNovo | 0.794 | 0.796 | 0.577 | 0.545 | 0.764 | 0.634 |
| | **PhysNovo** | 0.813 | 0.815 | 0.585 | 0.549 | 0.793 | 0.669 |
| Tomato | LIPNovo | 0.800 | 0.798 | 0.577 | 0.544 | 0.777 | 0.627 |
| | **PhysNovo** | 0.815 | 0.815 | 0.583 | 0.550 | 0.799 | 0.658 |
| **Mean** | LIPNovo | 0.804 | 0.804 | 0.595 | 0.565 | 0.790 | 0.661 |
| | **PhysNovo** | **0.820** | **0.820** | **0.601** | **0.572** | **0.816** | **0.692** |

*Table 4.* Comparison of parameter efficiency and task performance.

| Method | # Params | Amino Acid Level | | Peptide Level | |
|---|---|---|---|---|---|
| | | Prec. | Recall | Prec. | AUC |
| CasaNovo | 47.4M | 0.741 | 0.740 | 0.529 | 0.493 |
| LIPNovo | 68.4M | 0.797 | 0.797 | 0.582 | 0.547 |
| PhysNovo | 63.1M | **0.815** | **0.814** | **0.588** | **0.553** |

acid precision for the top 60% of predictions on the Nine-species dataset. This confirms that our iterative refinement mechanism successfully filters out uncertain hallucinations, yielding a highly calibrated model where confidence scores serve as a trustworthy proxy for physical validity.

**Robustness to Length Prediction.** A potential concern for NAR models is their reliance on the initial length prediction. However, analysis reveals that PhysNovo's standalone length predictor is highly accurate, with the predicted length distribution $P_\phi(L \mid c)$ sharply concentrated around the ground truth. Specifically, the predictor achieves an exact match rate of 94.2%, with 98.4% of predictions falling within ±1 residue, and 99.1% within ±2 residues.

To further evaluate the impact of length prediction errors on overall performance, we conduct a sensitivity analysis based on the absolute length error $\Delta L = |\hat{L} - L^*|$, as detailed in Table 6. An incorrect length prediction inevitably results in zero peptide-level precision due to the strict requirement of exact sequence matching. Because our discrete diffusion operates in a fixed-dimensional space, the model optimizes within the predicted length using MS1 and MS2 constraints rather than dynamically adjusting the length.

Importantly, Table 6 demonstrates that length errors lead to

*Table 5.* Comparison of inference speed and accuracy. PhysNovo enables a flexible speed–accuracy trade-off by adjusting the number of denoising steps ($T$).

| Method | Speed (spectra/s) | AA Prec. |
|---|---|---|
| CasaNovo (AR) | 11 | 0.697 |
| $\pi$-PrimeNovo (NAR) | **105** | 0.790 |
| **PhysNovo** ($T = 100$) | 46 | **0.815** |
| **PhysNovo** ($T = 50$) | 99 | 0.801 |

*Table 6.* Sensitivity analysis of PhysNovo's performance with respect to length prediction error $\Delta L = |\hat{L} - L^*|$. The length predictor is highly accurate (94.2% exact match), and the model exhibits graceful degradation in amino acid precision even when the predicted length is incorrect.

| Length Error ($\Delta L$) | Fraction | AA Prec. | Peptide Prec. |
|---|---|---|---|
| 0 (Exact match) | 94.2% | 0.839 | 0.624 |
| 1 | 4.2% | 0.531 | 0.000 |
| $\geq 2$ | 1.6% | 0.149 | 0.000 |

graceful degradation rather than catastrophic failure. For $\Delta L = 1$, which often arises from local isobaric substitutions (e.g., 'N' vs. 'GG'), the diffusion process still actively aligns the remainder of the sequence with the MS2 spectral evidence, recovering a substantial portion of the sequence (0.531 amino acid precision). Even for $\Delta L \geq 2$, terminal fragment alignment yields non-trivial precision (0.149). Ultimately, length prediction does not constitute a bottleneck for PhysNovo: errors affect less than 6% of samples. Even an oracle predictor with 100% exact match would only marginally improve the overall peptide-level precision (from 0.588 to 0.624), demonstrating the adequacy and reliability of our current module.

### 4.4. Ablation Study and Hyperparameter Tuning

**Component Ablation.** We evaluate the impact of physical mass (MS1) and global spectral (MS2) constraints on the Nine-species dataset (Table 7 and Figure 5). Removing the Hard Mass Constraint causes a notable drop in peptide precision ($0.588 \rightarrow 0.577$). Crucially, the Hard-only configuration (0.583) outperforms the Soft-only variant (0.577). This validates that global mass conservation serves as the fundamental boundary of the solution space; without the Knapsack kernel, the model struggles to ensure physical validity. The full model achieves the highest performance, confirming the synergy between strict mass constraint and spectral-guided refinement.

**Hyperparameter Tuning.** We investigate the sensitivity of the length prediction weight $\lambda$ and remasking threshold $\tau$ in Table 8. For $\lambda$, performance peaks at 0.5; excessively high weights distract the backbone, while lower weights

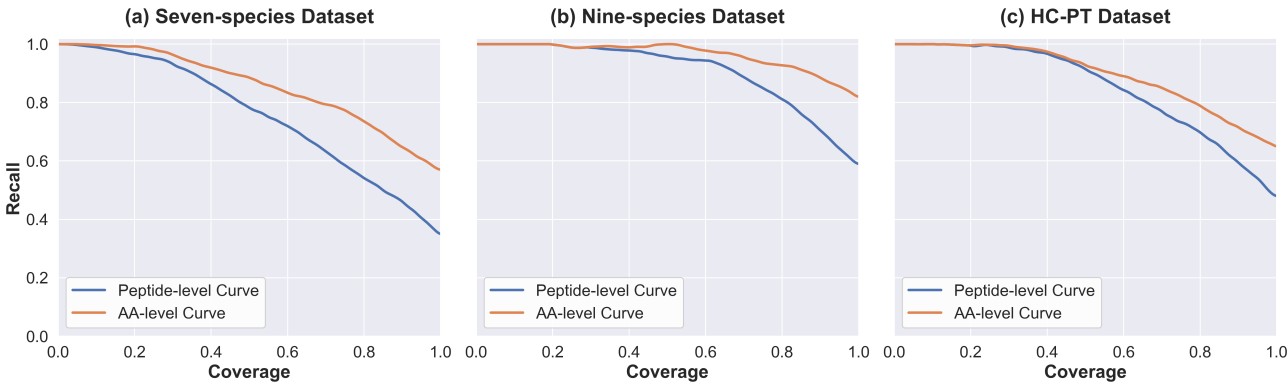

*Figure 4.* Recall-coverage curves illustrating the reliability of PhysNovo across three benchmarks. The curves depict the relationship between cumulative accuracy (y-axis, denoted as Recall) and coverage (x-axis) based on confidence ranking.

*Table 7.* Component ablation study on the Nine-species dataset. We evaluate the impact of Hard and Soft Constraints.

| ID | MS1 (Hard Constraint) | MS2 (Soft Constraint) | Amino Acid Level Prec. | Amino Acid Level Recall | Peptide Level Prec. | Peptide Level AUC |
|---|---|---|---|---|---|---|
| 1 | ✓ | ✗ | 0.806 | 0.806 | 0.583 | 0.551 |
| 2 | ✗ | ✓ | 0.795 | 0.796 | 0.577 | 0.540 |
| 3 | ✓ | ✓ | **0.815** | **0.814** | **0.588** | **0.553** |

*Table 8.* Sensitivity analysis of hyperparameters $\lambda$ and $\tau$. The default setting is highlighted.

| $\lambda$ | Amino Acid Precision | Peptide Precision | $\tau$ | Amino Acid Precision | Peptide Precision |
|---|---|---|---|---|---|
| 0.1 | 0.810 | 0.583 | 0.6 | 0.806 | 0.582 |
| **0.5** | **0.815** | **0.588** | **0.7** | **0.815** | **0.588** |
| 1.0 | 0.813 | 0.586 | 0.8 | 0.812 | 0.588 |
| 1.5 | 0.808 | 0.582 | 0.9 | 0.799 | 0.583 |

*Figure 5.* Ablation study on the Nine-species dataset. Comparison of PhysNovo against single-constraint variants across four metrics. Error bars denote standard deviation over 5 experiments.

tational cost is comparable to state-of-the-art AR baselines employing Beam Search ($L \times B \approx 75 \sim 150$ steps). More importantly, this design reflects a deliberate architectural trade-off: we actively invest computational resources into iterative reasoning to guarantee intrinsic physical validity during generation, thereby avoiding the costly post-hoc filtering required by unreliable one-shot predictions. A detailed theoretical analysis of how the computational complexity of our algorithm scales across broader and more challenging practical scenarios, including open search, complex PTMs, and isomers, is provided in Appendix F.

## 5. Conclusion

We introduce **PhysNovo**, a discrete diffusion framework that reframes peptide sequencing as physically constrained iterative reasoning. By integrating a knapsack-based mass constraint and spectral-guided error correction, PhysNovo overcomes the limitations of both sequential and one-shot decoding, ensuring strict mass conservation. Extensive experiments demonstrate that PhysNovo establishes a new SoTA across diverse benchmarks, exhibiting remarkable robustness on low-resolution spectra and superior generalization to unseen species. Beyond performance gains, our work highlights the critical value of embedding domain-specific physical constraints into deep learning architectures. We believe PhysNovo paves the way for more reliable, trustworthy proteomic discovery, particularly for identifying novel proteins and variants in open-world settings.

fail to guide length estimation. For $\tau$, the results favor a moderate threshold (0.7). Setting $\tau$ too high (0.9) causes a performance drop, confirming that over-aggressive remasking disrupts the semantic context required for diffusion.

**Computational Complexity and Trade-offs.** A natural concern regarding our iterative refinement process is the potential overhead of enforcing strict MS1 and MS2 constraints at every denoising step ($T = 100$). However, because our feasibility kernel relies entirely on continuous scalar arithmetic ($\mathcal{O}(1)$ per token), the marginal computational cost of these physical checks is virtually negligible compared to the diffusion process of the backbone. While executing $T = 100$ iterations inherently incurs higher latency than single-step (one-shot) NAR models, the compu-

## Acknowledgements

This research was supported in part by the Hong Kong Polytechnic University Internal Research Fund (P0057774, P0063303 through RIAIoT) and the Research Grants Council of Hong Kong's General Research Fund (Ref. No. 15208725).

## Impact Statement

This paper presents work whose goal is to advance the field of Machine Learning. There are many potential societal consequences of our work, none of which we feel must be specifically highlighted here.

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

# A. Details of Reasoning

---

**Algorithm 1** PhysNovo Inference: Physically Constrained Iterative Refinement

---

 1: **Input:** MS2 spectrum $\mathbf{x}$, Precursor info $\mathbf{c} = (m_{\text{prec}}, z)$, Diffusion steps $T$.
 2: **Hyperparameters:** Mass tolerance $\epsilon$, Remasking threshold $\tau$.
 3: **Output:** Predicted peptide sequence $\mathbf{y}_0$.
 4:   // 1. Physical Initialization (Sec. 3.1 & 3.3)
 5: Calculate target residue mass: $M_{\text{res}} \leftarrow (m_{\text{prec}} - m_{H^+}) \cdot z - w(\text{H}_2\text{O})$
 6: Predict sequence length: $\hat{L} \leftarrow \arg\max_L P_\phi(L \mid \mathbf{c})$
 7: Initialize noisy sequence: $\mathbf{y}_T \leftarrow [\text{MASK}]^{\hat{L}}$
 8:   // 2. Iterative Reasoning Process (Sec. 3.4)
 9: **for** $t = T, \dots, 1$ **do**
10:     // a. Neural Prediction with Spectral Conditioning
11:     Compute logits for denoised token $\mathbf{y}_0$: $\mathbf{l}_t \leftarrow f_\theta(\mathbf{y}_t, \mathbf{x}, \mathbf{c}, t)$
12:     // b. Hard Mass Constraint via Knapsack Kernel (MS1)
13:     **for** each position $j$ where $y_{t,j} = [\text{MASK}]$ **do**
14:       Calculate remaining mass budget $M_{\text{rem}}$ given fixed residues $\mathbf{y}_{\text{fixed}}$
15:       **for** each candidate amino acid $a \in \mathcal{V}$ **do**
16:         **if** NOT Reachable($M_{\text{rem}} - w(a)$, remaining_masks $-1$) **then**
17:           $l_{t,j}[a] \leftarrow -\infty$   // Prune invalid branch
18:         **end if**
19:       **end for**
20:     **end for**
21:     // c. Reverse Sampling
22:     Sample intermediate hypothesis $\tilde{\mathbf{y}}_{t-1}$ based on constrained logits $\mathbf{l}_t$
23:     // d. Global Error Correction (MS2)
24:     **for** each position $j$ **do**
25:       **if** Confidence($\tilde{y}_{t-1,j}$) $< \tau$ **then**
26:         $y_{t-1,j} \leftarrow [\text{MASK}]$   // Forced regression for re-reasoning
27:       **else**
28:         $y_{t-1,j} \leftarrow \tilde{y}_{t-1,j}$
29:       **end if**
30:     **end for**
31: **end for**
32: **return** $\mathbf{y}_0$

---

# B. Extended Analysis of Non-Autoregressive Baselines

In the main text, we discuss the limitations of emerging non-autoregressive (NAR) approaches, specifically noting their reliance on one-shot generation that often necessitates inefficient post-hoc filtering. Here, we provide a detailed analysis referencing the recent representative work, $\pi$-PrimeNovo (Zhang et al., 2025), to substantiate this claim.

**The Reliability Gap in One-Shot Generation.** Emerging NAR approaches in peptide sequencing, exemplified by $\pi$-PrimeNovo, typically employ a Transformer encoder-decoder architecture designed for parallel decoding (e.g., trained via Connectionist Temporal Classification loss). During inference, these models predict all amino acid probabilities simultaneously in a single forward pass. While this parallelization significantly accelerates inference, it inherently lacks an **iterative feedback mechanism**. If the neural network predicts a residue with high confidence that renders the total peptide mass inconsistent with the precursor mass (MS1), the model has no internal mechanism to revisit and correct this error within the generation pass.

**Dependence on Post-Hoc Filtering (PMC).** The authors of $\pi$-PrimeNovo acknowledge this limitation by introducing an external module termed **Precise Mass Control (PMC)**. PMC is essentially a post-hoc dynamic programming algorithm that searches for a mass-optimal path through the probability matrix generated by the neural network. Key evidence from their ablation study highlights the necessity of this post-processing:

- **Accuracy Dependency:** Removing the PMC module results in a significant performance drop (approximately 7% in peptide recall), indicating that the raw one-shot output of the neural network is frequently physically invalid or unreliable.
- **Efficiency Trade-off:** While the pure neural network is extremely fast, the inclusion of the PMC post-processing step increases inference latency by approximately $3.7\times$.

**Contrast with PhysNovo.** This analysis underscores our argument: existing NAR models treat physical constraints as an afterthought (post-hoc filtering) rather than an intrinsic generative rule. In contrast, PhysNovo integrates the Knapsack Hard Mass Constraint directly into the iterative diffusion steps. By pruning the search space *during* generation, PhysNovo ensures that the final output is physically valid by design. Unlike methods that rely on external solvers to fix unreliable predictions, our approach achieves intrinsic validity through iterative reasoning, representing a fundamental shift in how physical laws are integrated into deep learning.

## C. Length Prediction Details

In Section 3.3, we formulated peptide length prediction as a classification problem. Here, we provide the specific configurations and inference logic used in our experiments.

**Range Configuration ($L_{\min}, L_{\max}$).** We define the plausible range of peptide lengths as $[L_{\min}, L_{\max}] = [7, 40]$. This range is selected based on the statistical distribution of enzymatic digestion products (e.g., Trypsin/Lys-C) in the Swiss-Prot database. Peptides shorter than 7 amino acids are typically not unique enough for reliable protein identification, while peptides longer than 40 residues are rare in standard bottom-up proteomics workflows and often suffer from poor fragmentation efficiency.

**Network Architecture.** The length predictor $f_\phi$ is a 3-layer MLP taking the normalized precursor mass and charge as input. The output dimension is $C = L_{\max} - L_{\min} + 1 = 34$ classes. The mapping from class index $i \in [0, 33]$ to physical length is given by $L = i + L_{\min}$.

**Inference Logic.** During the inference stage, we determine the target length $\hat{L}$ by selecting the class with the highest probability. To avoid confusion with the position index $l$ used in the main text (Eq. 2), we denote the candidate length as $k$:

$$\hat{L} = \mathrm{argmax}_{k \in [L_{\min}, L_{\max}]} P_\phi(L = k \mid \mathbf{c}). \tag{12}$$

Once $\hat{L}$ is determined, we construct the initial noise tensor $\mathbf{y}_T \in \mathbb{R}^{\hat{L} \times |\mathcal{V}|}$. Unlike autoregressive models that can dynamically determine the end-of-sequence token, discrete diffusion models require a fixed tensor shape throughout the denoising trajectory. Therefore, the accuracy of this prediction is critical. Empirically, we find that the strong correlation between precursor mass and sequence length allows the MLP to achieve high accuracy, providing a solid foundation for the subsequent diffusion process.

## D. Datasets & Evaluation Metrics and Implementation Details

### D.1. Pretraining Data Generation Details

To construct the unlabeled peptide corpus, we utilized the **UniProtKB/Swiss-Prot** database (Release 2024_01, containing 571,768 entries). The *in silico* digestion was performed using the Pyteomics library with the following strict parameters to ensure reproducibility:

- **Enzymes**: Trypsin (cleaves at K, R except before P) and Lys-C (cleaves at K).
- **Missed Cleavages**: Allowed up to **2** missed cleavage sites per peptide, simulating incomplete digestion commonly observed in shotgun proteomics.
- **Length Constraint**: Peptides were filtered to retain lengths between 7 and 40 amino acids (inclusive).
- **Filters**: We excluded sequences containing non-canonical amino acids (B, J, O, U, X, Z) to align with the standard vocabulary.

After removing duplicates, this process yielded the final corpus of 15,535,514 unique peptide sequences.

### D.2. Benchmark Dataset

We evaluate PhysNovo on the following three benchmarks, covering a diverse range of spectral qualities and biological contexts:

*Table 9.* Statistics of the benchmark datasets.

| Dataset | Avg. Peptide Length | PTM Class | Train / Valid / Test Size |
|---|---|---|---|
| Seven-species | 15.79 | 3 | 317,009 / 17,740 / 17,094 |
| Nine-species | 15.01 | 3 | 499,402 / 28,572 / 27,142 |
| HC-PT | 12.53 | 1 | 213,284 / 25,718 / 26,536 |

- **Nine-species** (Tran et al., 2017): A high-resolution dataset extensively used in prior research. It contains spectra from nine distinct organisms and is notable for incorporating three common PTMs: Methionine oxidation, as well as Asparagine and Glutamine deamidation. We strictly follow the cross-species protocol, using Saccharomyces cerevisiae (yeast) exclusively for testing to assess generalization to unseen species.
- **Seven-species** (Tran et al., 2017): A low-resolution counterpart derived from seven species. This benchmark evaluates robustness against lower-quality spectral data. Consistent with the Nine-species setup, our evaluation focuses on the held-out yeast species.
- **HC-PT** (Eloff et al., 2023): A high-resolution dataset of human-origin peptides. It consists of synthetic tryptic peptides representing all canonical human proteins and their isoforms, alongside peptides generated by alternative proteases and HLA peptides.

### D.3. Evaluation Metrics

This section delineates the detailed definitions and calculation protocols for the evaluation metrics employed in our experiments. In all evaluations, Leucine (L) and Isoleucine (I) are treated as equivalent due to their isomeric nature and indistinguishable mass.

**Peptide-level Precision.** This metric serves as the principal indicator of a model's practical utility, as the ultimate goal of *de novo* peptide sequencing is to assign complete and accurate peptide sequences to each mass spectrum. A predicted peptide $\mathbf{y}$ is considered correct only if its entire amino acid sequence matches the ground truth $\hat{\mathbf{y}}$ (i.e., $\mathbf{y} = \hat{\mathbf{y}}$). Given a dataset of $N_{all}^p$ spectra, if a model correctly predicts $N_{match}^p$ peptides, the peptide-level precision is calculated as:

$$\text{Peptide Precision} = \frac{N_{match}^p}{N_{all}^p} \tag{13}$$

**Peptide-level AUC.** The Area Under the Precision-Recall Curve (AUC) provides a comprehensive assessment of the model's performance across different confidence thresholds. The calculation process involves the following steps:

1. **Confidence Scoring**: Define the confidence score $sc(\mathbf{y})$ of a peptide $\mathbf{y}$ as the mean of its constituent amino acid confidence scores (predicted probabilities).

2. **Ranking**: Sort all predicted peptides in the test set in descending order based on their confidence scores $sc(\mathbf{y})$.

3. **Integration**: Starting from the highest confidence prediction, iteratively calculate the cumulative precision and recall values. These values serve as the y-axis and x-axis, respectively, to plot the precision-recall curve. The AUC is obtained by calculating the area under this curve.

A higher AUC indicates that the model is better at assigning higher confidence scores to correct predictions, which is crucial for distinguishing reliable identifications in real-world applications.

**Amino Acid-level Precision and Recall.** These metrics offer a granular assessment of reconstruction quality. An individual predicted amino acid is considered a **match** if it satisfies two criteria:

1. **Mass Accuracy**: The mass difference between the predicted residue and the ground truth residue is $< 0.1$ Da.

2. **Position Accuracy**: The mass of the prefix (N-terminal accumulation) or suffix (C-terminal accumulation) at this position differs by $\leq 0.5$ Da from the ground truth.

*Table 10.* Monoisotopic masses of amino acid residues and PTMs used in PhysNovo. The values correspond to the residue mass (without N-terminal H and C-terminal OH).

| Residue | Mass (Da) | Residue | Mass (Da) |
|---|---|---|---|
| Alanine (A) | 71.03711 | Leucine (L) | 113.08406 |
| Arginine (R) | 156.10111 | Lysine (K) | 128.09496 |
| Asparagine (N) | 114.04293 | Methionine (M) | 131.04049 |
| Aspartic Acid (D) | 115.02694 | Phenylalanine (F) | 147.06841 |
| Cysteine (C)* | 160.03065 | Proline (P) | 97.05276 |
| Glutamine (Q) | 128.05858 | Serine (S) | 87.03203 |
| Glutamic Acid (E) | 129.04259 | Threonine (T) | 101.04768 |
| Glycine (G) | 57.02146 | Tryptophan (W) | 186.07931 |
| Histidine (H) | 137.05891 | Tyrosine (Y) | 163.06333 |
| Isoleucine (I) | 113.08406 | Valine (V) | 99.06841 |
| **Post-Translational Modifications (Variable)** | | | |
| Oxidation (M) | 147.03540 | Deamidation (N) | 115.02695 |
| Deamidation (Q) | 129.04260 | | |

Cysteine mass includes fixed Carbamidomethylation (+57.02146).

Let $N_{match}^{aa}$ be the number of matched amino acids, $N_{pred}^{aa}$ be the total number of predicted amino acids, and $N_{truth}^{aa}$ be the total number of amino acids in the ground truth sequences. The metrics are defined as:

$$\text{AA Precision} = \frac{N_{match}^{aa}}{N_{pred}^{aa}}, \quad \text{AA Recall} = \frac{N_{match}^{aa}}{N_{truth}^{aa}} \tag{14}$$

**PTM-level Precision and Recall.** Post-translational modifications (PTMs) are critical for protein function but are challenging to identify due to subtle mass shifts. Similar to the amino acid-level metrics, we define $N_{match}^{ptm}$ as the number of correctly identified PTMs (where both the residue type and the modification type are correct, and position criteria are met). Let $N_{pred}^{ptm}$ be the total number of predicted PTMs and $N_{orig}^{ptm}$ be the total number of PTMs in the ground truth. The metrics are formulated as:

$$\text{PTM Precision} = \frac{N_{match}^{ptm}}{N_{pred}^{ptm}}, \quad \text{PTM Recall} = \frac{N_{match}^{ptm}}{N_{orig}^{ptm}} \tag{15}$$

### D.4. Implementation Details

**Model Architecture.** The PhysNovo backbone $\Phi_\theta$ employs a 12-layer Bidirectional Transformer architecture. We set the hidden dimension to $d = 512$ and the feed-forward network (FFN) dimension to 2048, with 16 attention heads per layer. To integrate spectral evidence, we inject the peak embeddings into the backbone via multi-head cross-attention modules inserted after the self-attention block in each layer. The PeakEncoder projects the mass and intensity features into the $d$-dimensional vector space using sinusoidal functions followed by a linear transformation. The length prediction head $f_\phi$ consists of a 3-layer MLP with a hidden size of $512$ and a dropout rate of $0.1$.

**Inputs and Vocabulary.** The residue vocabulary $\mathcal{V}$ consists of 23 tokens: the 20 canonical amino acids and 3 common post-translational modifications (Oxidation of Methionine, Deamidation of Asparagine and Glutamine). For the discrete diffusion process, we augment this vocabulary with two special tokens: [PAD] for batch alignment and [MASK] for the corruption process, resulting in a total model vocabulary size of 25. We filter the input MS/MS spectrum to retain the top $N = 150$ peaks based on intensity. The maximum peptide length is set to $L_{\max} = 40$.

**Training Protocol.** (1) *Pretraining:* The model is first trained unconditionally on the *in silico* digested Swiss-Prot corpus to learn the generative prior of valid peptide sequences. We train for 30 epochs using the AdamW optimizer with a learning rate of $5 \times 10^{-4}$ and a weight decay of $0.01$. The learning rate follows a linear warm-up for the first 2% of steps, followed by a cosine decay. (2) *MS/MS-Constrained Fine-Tuning:* The model is then fine-tuned on the training splits of the spectral benchmarks (Sec 4.1). We jointly optimize the diffusion loss and length prediction loss (Eq. (9)) with $\lambda = 0.5$. During this stage, the learning rate is reduced to $5 \times 10^{-5}$, and training proceeds for 50 epochs with a batch size of 64. Gradient

clipping with a norm threshold of 1.0 is applied to stabilize optimization.

**Inference Configuration.** During inference, we perform iterative denoising over $T = 100$ discrete timesteps. The knapsack-based feasibility kernel enforces the precursor mass constraint with a tolerance of $\epsilon = 0.02$ Da (approximately 20 ppm). For global error correction, we apply a confidence threshold of $\tau = 0.7$ to identify and regenerate low-confidence residues. All experiments were conducted on a single NVIDIA GeForce RTX 4090 GPU.

**Physical Constants and Mass Definitions.** The knapsack-based feasibility kernel strictly enforces mass conservation based on precise monoisotopic masses. To ensure reproducibility, we list the exact mass values assigned to each token in the vocabulary $\mathcal{V}$ in Table 10. Note that Cysteine (C) is treated with a fixed Carbamidomethylation modification (+57.02146 Da) following standard proteomics protocols. The masses of the proton (1.00728 Da) and water (18.01056 Da) are used to derive the target residue mass $M_{\text{res}}$.

## E. Discussion on Train-Inference Consistency

A potential theoretical concern regarding the Global Error Correction strategy is the alignment between the training noise distribution and the adaptive remasking during inference. During training, the diffusion backbone learns to denoise sequences where masked positions are sampled uniformly according to a fixed variance schedule $\beta_t$. During inference, however, we adaptively force low-confidence tokens (below $\tau$) back to [MASK], potentially altering the effective noise level at timestep $t$.

We justify the validity of this strategy from three perspectives:

1. **Model Robustness:** The backbone $\Phi_\theta$ is essentially a conditional Masked Language Model trained on a vast corpus. Empirical studies in discrete diffusion (Austin et al., 2021) suggest that such models are robust to variations in mask density, as the timestep embedding primarily acts as a gating mechanism for the temperature of the output distribution rather than a strict counter of masked tokens.
2. **Minimal Deviation:** Empirically, with the optimal threshold $\tau = 0.7$, the remasking operation typically affects only a small fraction of residues in the later denoising stages. Thus, the actual signal-to-noise ratio remains close to the schedule, preventing significant distributional shift.
3. **Refinement as Dynamics:** The remasking step can be interpreted as a discrete analogue to corrector steps in continuous diffusion score-based models. By rejecting low-confidence predictions and reverting them to the neutral [MASK] state, we allow the model to re-sample these positions conditioned on the updated, higher-quality context, guiding the trajectory back toward the high-probability manifold of valid peptides.

## F. Scalability to Open Search, Complex PTMs, and Isomers

As discussed in the main text, ensuring physical validity via MS1 and MS2 constraints introduces a deliberate trade-off in inference efficiency. In this section, we provide a detailed analysis of how the computational complexity of our proposed algorithm scales when extended to broader and more challenging practical scenarios.

**Core Computational Complexity.** A potential concern regarding the physical constraints is the risk of combinatorial explosion. However, the knapsack-based feasibility kernel in PhysNovo does not exhibit exponential complexity in practice. Because the bounded reachability reduces to a continuous $\mathcal{O}(1)$ scalar check per token (as formulated in Eq. 11), the overall worst-case computational complexity for a peptide of length $L$ over $T$ denoising steps is strictly bounded by $\mathcal{O}(T \cdot L \cdot |\mathcal{V}|)$, where $|\mathcal{V}|$ is the vocabulary size. This boundary holds because the check depends solely on global mass bounds ($w_{\min}, w_{\max}$) rather than exhaustive sequence enumeration. Consequently, expanding the vocabulary to accommodate modifications incurs only a linear scaling in complexity.

**Scaling to Open Search and Complex PTMs.** While the linear scaling property guarantees fundamental tractability, extreme large-scale PTM settings and Open Search scenarios inevitably expand the vocabulary size $|\mathcal{V}|$ significantly, introducing additional computational overhead. To maintain optimal practical efficiency under these settings, our framework is compatible with several optimization strategies:

1. **Mass-grouped Compression:** Grouping modifications with identical or near-identical mass shifts to reduce the effective vocabulary size $|\mathcal{V}|$ during the diffusion steps.
2. **Cache Reuse:** Reusing the calculated feasibility cache across adjacent denoising steps, exploiting the fact that the underlying mass constraints remain static.

3. **Broader Pretraining Prior:** Expanding the pretraining corpus to encompass a broader spectrum of PTMs. This further concentrates the learned probability mass of the denoising network on physically plausible candidates, effectively shrinking the active search space.

**Resolving Isomers.** In the context of isomeric or isobaric peptides, the complexity is not strictly computational but rather informational. In PhysNovo, the MS1 constraint serves as an efficient zero-overhead feasibility filter to discard invalid mass combinations, while the final sequence discrimination relies entirely on the MS2-conditioned likelihood. This complementary design enables the model to organically resolve isomeric ambiguities utilizing fragment ion evidence. We acknowledge that fundamentally indistinguishable isomers (e.g., Leucine and Isoleucine) remain an inherent limitation of mass spectrometry itself; nevertheless, our framework maximizes the utilization of available spectral evidence without introducing exponential computational branching.

