# OpenReview forum: "Discrete Diffusion with Physical Mass Constraints for \emph{De Novo} Peptide Sequencing"
_ICML.cc/2026/Conference — ICML 2026 regular_

### Official Review · Reviewer_Zpzj · 2026-02-13

**Soundness:** 4
**Presentation:** 3
**Significance:** 3
**Originality:** 2
**Overall Recommendation:** 5
**Confidence:** 2

**Summary:**

**Following the rebuttal period, I have upgraded the score from a 4 to a 5. Some of the authors' claims relied on information I could not verify independently, leading me to lower my confidence score. While I do not believe this work is a solution to the problem, I believe it will be interesting to practitioners in the field, and raises interest in potential extensions for researchers.**

The presented problem domain is the determination of amino acid sequences from MS/MS data; a very high-impact problem in the field. Previous techniques often work with autoregressive germination or direct prediction. They introduce a technique, PhysNovo, which uses global reasoning and iterative refinement to guide discrete diffusion. They further use a feasibility kernel to enforce mass conservation throughout the generation. This incurs a very high computational cost, but it is argued that this is worth the time in its ability to solve generation without further post-processing.

The key contributions are a technique that follows (some of the) key physical laws, and allows for consideration of the while sequence at once.

Consistent improvements are shown across datasets when compared with a variety of existing techniques. The performance gains are often relatively small, but consistent across the tests.

The main concerns raised are: the np-hardness of the knapsack problem, the reliance on L^* predictions, the high computational cost of the model, and the lack of clarity around the padding and masking terms. These however, are outweighed by the consistency of the performance gains, including substantial gains in some tests, as well as the argument that this technique could inspire further development and research into the integration of physical conservation constraints into the diffusion process in this, or related, fields.

**Compliance With Llm Reviewing Policy:**

Affirmed.

**Final Justification:**

Following the rebuttal period, I have revised my score from 4 to 5, but lowered my confidence rating. Some of the authors’ claims rely on information I was unable to independently verify, which is what lowers my confidence. I also continue to have reservations about the magnitude of the reported performance gains, as they remain seemingly quite small.

The paper’s primary contribution is not the absolute improvement in performance, but the demonstration that an alternative technique can achieve comparable results on this problem. While the technique itself is not novel and has been used throughout ML literature, its application to this problem is new. This work shows that an existing machine learning approach can match the performance of current methods in the literature.

I initially found it difficult to choose between a 4 and a 5. This topic is tangential to my area of research, which makes it challenging to verify the impact claims made during the rebuttal. However, assuming those claims hold, I believe the paper meets the bar for a score of 5.

**Key Questions For Authors:**

1 - Reasoning behind the use of a 12-layer architecture is not given… are there empirical results supporting that this is a strong choice of model size?

2 - Why are sequences padding to the maximum length? A Transformer encoder is inherently capable of working on variable length inputs. Is this just in reference to the runtime batching performance? If so, this is standard practice and usually omitted in discussions of architecture design. Similar comments go for the attention mask. Is this included for the benefit of the diffusion process?

3 - Does the use of the iterative process ever lead to a degenerative or cyclic sequence of updates?

**Limitations:**

1 - Initial guess of the sequence length is quite important, as failure to get it correct will guarantee incorrect results. There is no correction mechanism for this component. Incorrect predictions result in potentially wasting a very costly generation process. The only mitigating factor to this concern is the experimental results.

2 - Across the total of 12 metrics, the performance gains are quite small on 4 of the reported scores, and worse on 1.

3 - The core idea of the model, constrained diffusion, is not necessarily new. Given this is an application paper and the exact technique has not been used, I do not weigh this as a substantial concern.

4 - The results themselves may not justify a wholesale change to the established workflows (depending on difficulty of interchanging techniques), but it raises strong arguments that this a meaningful direction of research.

**Strengths And Weaknesses:**

Strengths

1 - Argument for the need of mass conservation laws is well formulated.

2 - The use of kernel restraints in diffusion is not in itself, novel, but this particular kernel has not been used before in literature to the best of my knowledge. This makes it a substantial new contribution.

3 - The selection of the Sinusoidal PeakEncoder is well justified.

4 - The reported performance of the model shows it consistently outperforms existing methods. The selection of many testing domains and the number of existing models is substantial.

5 - The component ablation studies are very helpful in justifying assertions about what leads to improved model performance.

6 - (small note) the graphics and charts are very well made.

I do not consider the small reduction in the number of parameters to be a meaningful strength… the reduction is relatively small and not significant enough to make a difference on modern computing setups. PhysNovo still has substantially more parameters than CasaNovo.


Weaknesses

1 - Autoregressive decoding does allow for context from the whole input sequence to be considered at once, as well as context from the whole sequence generated thus far to be considered. The only limitation is that future tokens cannot effect the prediction of past tokens. This is not clearly delineated in the work, which often is written in such a way that it would imply that the consideration of the prior context is limited.

2 - The knapsack problem is np-hard… this paper does not address potential failure cases from the approximation used. Does this create invalid structures, or is it just a worse, but still physically valid, result?

3 - The performance gains are noticeable, but relatively small.

4 - Studies on the ability of the model to predict L^* are not given, despite the importance of this value to the generation process.

5 - As acknowledged in the paper, the time burden to run one generation is quite high. In physical sciences fields this is often an easily justifiable tradeoff, making this a relatively minor limitation.

---

> ### Author Rebuttal · Authors · 2026-03-30
>
> We thank the reviewer for the insightful feedback. We address your concerns below, and will include all these discussions in the final revision.
>
> **Clarification on AR. (W1)**
> We appreciate this clarification and did not intend to suggest that AR models ignore the full spectrum or prefix context. Our point concerns the inherent limitation of the sequence-to-sequence translation paradigm: left-to-right generation causes early errors to propagate and degrade later predictions, while lacking an effective mechanism to revise past decisions during generation.
>
> **Knapsack and Physical Validity. (W2)**
> The knapsack approximation does not introduce MS1-violating candidates. It acts strictly as a feasibility filter: all retained sequences satisfy the precursor mass within tolerance. We note that mass consistency alone does not fully determine physical validity. In our framework, structural plausibility is primarily enforced by MS2 conditioning and the diffusion refinement process, where MS2 provides sequence-level global constraints. This reduces the likelihood that approximation errors propagate into degenerate solutions. Overall, the knapsack kernel serves to prune infeasible candidates, the approximation may lead to suboptimal (but still mass-consistent) predictions; however, by combining MS1 constraints with MS2-based global reasoning, the model typically produces mass-consistent and structurally plausible peptide sequences in practice (Details in Reviewer 2yH5--(Q3)).
>
> **Significance of Performance Gains. (W3, L2)**
> Although some gains appear small, *de novo* sequencing evaluates exact full-sequence matches; due to the exponential sequence space, even 1–2% absolute improvement represents a substantial improvement in recovering usable proteins. More importantly, PhysNovo shows substantial gains in PTM identification: on the Seven-species dataset, it outperforms LIPNovo by +5.0% prec. and +5.1% recall, with consistent improvements (e.g., +3.1% prec. on HC-PT). These results highlight its strong sensitivity to subtle mass shifts and its effectiveness in handling complex, real-world PTMs.
>
> **Length Prediction. (W4, L1)**
> See our discussion for **Reviewer jiuq--(W1, W2, Q1)**.
>
> **Computational Trade-Off. (W5)**
> See our discussion for **Reviewer jiuq--(Q3)**.
>
> **12-Layer. (Q1)**
> We selected a 12-layer architecture to balance capacity and efficiency, and aligning with standard protein language models and recent methods like LIPNovo for fair comparisons. To validate this choice, we scaled CasaNovo from 9 layers (47.4M) to 12 layers (69.4M), comparable to our model. As shown in Table, this yields only marginal gains (+0.9% AA prec., +1.1% AA recall), indicating that increased depth alone does not drive performance. In contrast, our 12-layer PhysNovo (63.1M) significantly outperforms all baselines, showing that improvements stem from architectural design rather than scale. A larger variant (~150M, 30 layers) provides limited additional gains (+0.8% peptide prec.) at substantially higher cost due to diffusion. Thus, the 12-layer model offers a strong Pareto trade-off between accuracy and efficiency for large-scale proteomics.
> |Method|Params|AA-Level Prec.|AA-Level Recall|Peptide Level Prec.|Peptide Level AUC|
> |:-|:-|:-:|:-:|:-:|:-:|
> |CasaNovo (Retrain Result)|47.4M|0.741|0.740|0.529|0.493|
> |CasaNovo (Extended)|69.4M|0.750|0.751|0.539|0.494|
> |LIPNovo|68.4M|0.797|0.797|0.582|0.547|
> |**PhysNovo (12-layer)**|**63.1M**|**0.815**|**0.814** |**0.588**|**0.553**|
> |PhysNovo (~150M)|152.4M|0.832|0.830|0.596|0.562|
>
> **Padding and Masking. (Q2)**
> We agree that padding facilitates efficient batching, but in discrete diffusion it also enforces a consistent state dimension $L_{\max}$. This is critical, as diffusion operates over a fixed-dimensional Markov trajectory; varying lengths would force the model to learn multiple, inconsistent diffusion processes, harming training stability. Attention masking is essential to exclude padded positions from both self-attention and the diffusion objective. Since diffusion performs global iterative refinement, unmasked padding would introduce spurious signals, interfering with spectral and mass-constrained alignment. At inference, we predict length $\hat{L}$ and initialize noise at that exact size, avoiding padding.
>
> **Degenerative & Cyclic Updates. (Q3)**
> We do not observe degenerative or cyclic behavior, both theoretically and empirically, for three reasons. First, the directed diffusion schedule ($T=100$) monotonically increases the signal-to-noise ratio, ensuring convergence toward a stable solution. Second, confidence-based remasking acts as annealing: as confidence increases, remasking decreases, preventing repeated token flipping. Third, the MS1 knapsack kernel deterministically prunes infeasible candidates, anchoring the trajectory rather than inducing oscillations. Empirically, token updates stabilize as $t \rightarrow 0$.
>
> **Novelty. (L3)**
> See our discussion for **Reviewer 6dbg--(W3)**.

---

> > ### Author Rebuttal · Reviewer_Zpzj · 2026-03-31
> >
> > Contesting the authors’ assertions in some statements is beyond my knowledge and my ability to effectively research in this time frame. Accepting their responses yields a clear score of “5 - Accept” for this submission, although I must lower my confidence score to account for this uncertainty. Previously I had given a rating of: "4 - Weak Accept" with confidence 3.
> >
> > The main concerns I have remaining are:
> >
> > 1 - The results do not show large performance gains, and even show some failure cases in practice. Not being a practitioner in the field who uses this technology, I find it hard to determine if the increased compute cost is worth the slight performance gain. I do maintain concerns regarding if practitioners will find the performance gains sufficient to justify upending their established workflows. I acknowledge the authors’ comments in the rebuttal, and will accept their justification. -> accept, although with lower confidence score.
> >
> > 2 - The algorithm is not new in and of itself, although it is new to the field -> weak accept
> >
> > **Clarification on AR:**
> >
> > I appreciate the explanation and I am satisfied with this response! Assuming minor clarity updates are made to the submission I have no further concerns!
> >
> > **Knapsack and Physical Validity:**
> >
> > Does this filtering remove potentially correct candidates as well? How is that accounted for by the algorithm? Or, should we accept that we may lose some viable candidates and rely on the experimental results to show that this is not a concern?
> >
> > **Significance of Performance Gains:**:
> >
> > I acknowledge the authors’ statements regarding the importance of the slight performance gains to practitioners in the field. I have checked into this independently as much as is reasonably feasible, but I cannot confirm their statements with certainty. The decrease in the exponential growth does result in a substantial gross decrease, although my intuition would day that the remaining error is still very large and sufficient to create substantial issues for practitioners. This can be viewed as a limitation of the SOA in the field, rather than this algorithm however.
> >
> > **Length Prediction:**
> >
> > The only new thing from this response is the statement that failures do not lead to catastrophic results. In my initial response I noted the strong experimental justification despite the lack of theoretical guarantees. While this would make me uncomfortable on a personal level, I do not want to overemphasize my personal preference for theoretical guarantees over experimental justification, and will thus not hold back the paper significantly. *I do believe that the authors should certainly include in the main body of the paper the statement about the graceful degradation rather than catastrophic failure.*
> >
> > **Computational Tradeoff:**
> >
> > I see the argument that this method can be set to run in the same time/compute budget as existing works and still produce (slightly) superior performance. This fully satisfies my concern, since the technique is useable under the same constraints of existing literature/techniques. Additionally, I note that it is still trainable on hardware that is feasible for many researchers to acquire.
> >
> > **12-Layer:**
> >
> > While not an ideal testing scenario (ideal might be testing a variety of sizes and reporting a table of results), I understand there are often limitations on compute available that render such experiments infeasible, and I am satisfied with the provided justification. Assuming greater uptake in the community, optimized model sizes can be inferred from the body of work in aggregate.
> >
> > The table is helpful, but does raise another question: “Why is PhysNovo (150M) not selected?” It demonstrates the strongest performance. Is the selection of the 12-layer over the 150M to do with the performance vs compute tradeoff?
> >
> > **Padding and Masking:**
> >
> > This response is well reasoned and satisfies my concerns regarding this point. I do not anticipate this being added to the main body of the paper, but a note to a section in the Appendix would be useful for people like myself who are tangentially related to this work.
> >
> > **Degenerative and Cyclic Updates:**
> >
> > I appreciate the extra information, and it satisfies my concern. Similarly to the padding and masking section, I hope that the authors will consider adding further explanation in the paper or related works that will help unfamiliar readers.
> >
> > **Novelty:**
> >
> > I appreciate the authors’ viewpoint regarding this. I believe that we have agreement that the technique is not itself fundamentally new, but it is new to apply it to this domain and use it in such a manner. While a weaker contribution than creating a totally new algorithm, given that this is in an “Applications -> Health and Medicine” track, I believe that this work satisfies the novelty constraints of this track.

---

> > > ### Author Response · Authors · 2026-04-02
> > >
> > > We sincerely thank the reviewer for the detailed follow-up and for the constructive evaluation. We are encouraged that most concerns have been addressed, and we further clarify the remaining points below.
> > >
> > > **Clarification on AR:**
> > > We thank the reviewer for the positive feedback. We will incorporate clarity improvements in the revised version to further improve the presentation and avoid potential misunderstandings.
> > >
> > > **Knapsack and Physical Validity:**
> > > We thank the reviewer for the insightful question. In our case, the MS1 knapsack constraint is designed to be conservative (within instrument-aligned tolerance), and serves primarily as a feasibility filter rather than aggressively pruning plausible candidates.
> > > The results in **Tab. 1** show that the primary role of the MS1 constraint is to eliminate invalid candidates. Without MS1, 18.4% of incorrect predictions violate mass conservation, with large deviations (~117 Da), indicating hallucinated residue compositions. In contrast, the full model enforces strict mass consistency (0.0% violation), ensuring all generated sequences remain physically valid.
> > >
> > > **Table 1**
> > > |Model Configuration|Peptide Prec.|Mass Violation Rate| Degree of Violation (Da)|
> > > |:-|:-:|:-:|:-:|
> > > |PhysNovo (Full Model)|0.588|0.0%|0.0|
> > > |PhysNovo (Soft-only)|0.577|18.4% (In Incorrect Predictions)|$\approx$ 117.1 ($\approx$ 1 residue)|
> > >
> > > To further analyze whether the MS1 and MS2 constraints may discard otherwise correct candidates, we perform a comparison across two variants: **MS1-only**, and **MS2-only (soft-only)** (**Tab. 2**), where correctness is defined based on exact peptide-level match. Specifically, cases where the MS1 constraint may discard an otherwise valid candidate (i.e., MS2-only succeeds but the full model fails) are exceedingly rare (≈0.5%). Similarly, cases where adding MS2 information misleads an otherwise correct MS1 prediction are also minimal (≈0.7%). This indicates that while a minuscule fraction of viable candidates may be lost, the filtering mechanism overwhelmingly preserves correct candidates and is largely complementary.
> > >
> > > **Table 2**
> > > | Case | Proportion | Interpretation |
> > > | :--- | :--- | :--- |
> > > | MS1 Correct → Full Incorrect | ≈0.7% | Rare interference |
> > > | MS2 Correct → Full Incorrect | ≈0.5% | Rare over-constraining |
> > >
> > > Furthermore, as illustrated in Sec. 4.3 (Fig. 4), the knapsack constraint acts as a guiding mechanism during diffusion, progressively steering candidates toward mass-consistent regions rather than aggressively pruning plausible ones. This enables global refinement under physical constraints, instead of enforcing hard decisions that could discard valid sequences. Therefore, while theoretical edge cases may exist (e.g., due to measurement noise), both quantitative and qualitative evidence indicate that the MS1 constraint primarily removes infeasible solutions with negligible impact on valid predictions in practice.
> > >
> > > **Significance of Performance Gains:**
> > > We thank the reviewer for the careful consideration and for independently examining this point. Building on the current framework, we will continue to explore more accurate and effective approaches within this paradigm.
> > >
> > > **Length Prediction:**
> > > We thank the reviewer for the thoughtful assessment and helpful suggestion. We will explicitly incorporate the statement of graceful degradation (rather than catastrophic failure) into the main body of the paper, and refine the related discussion for clarity and precision.
> > >
> > > **Computational Tradeoff:**
> > > We sincerely thank the reviewer for the positive assessment and for recognizing the practical value of our method.
> > >
> > > **12-Layer:**
> > > We thank the reviewer for the positive assessment and the follow-up question. Yes, this choice is primarily driven by the performance–compute trade-off. While the 150M model achieves the strongest performance, it incurs substantially higher computational cost, particularly due to the iterative nature of diffusion at inference time. In contrast, the 12-layer configuration provides a strong performance–efficiency balance while remaining comparable in scale to existing state-of-the-art methods. This allows for fair comparison and, more importantly, demonstrates that the gains of PhysNovo stem from architectural design rather than increased model size.
> > >
> > > **Padding and Masking & Degenerative and Cyclic Updates:**
> > > We thank the reviewer for the suggestion. We will include additional clarification on "Padding and Masking" and "Degenerative and Cyclic Updates" in the Appendix of the revised version to improve readability for a broader audience.
> > >
> > > **Novelty:**
> > > We thank the reviewer for the thoughtful assessment and positive evaluation.

---

### Official Review · Reviewer_6dbg · 2026-02-19

**Soundness:** 3
**Presentation:** 3
**Significance:** 3
**Originality:** 3
**Overall Recommendation:** 5
**Confidence:** 3

**Summary:**

This paper explores the fundamental task of de novo peptide sequencing, which involves reconstructing amino acid sequences directly from tandem mass spectrometry data without relying on reference peptide databases. The study aims to overcome the limitations of current Auto-Regressive and Non-Auto-Regressive models, which often suffer from error propagation or fail to strictly adhere to physical mass constraints. To address this, the authors propose PhysNovo, a discrete diffusion-based framework that reformulates sequencing as an iterative, physically constrained inference process rather than a simple sequence-to-sequence translation task. The framework consists of three core phases: 1. Structure-aware pre-training on in silico digested peptide sequences to learn biochemical priors. 2. MS/MS-constrained fine-tuning, which integrates soft spectral alignment and precursor mass conditioning into the diffusion backbone. 3. Iterative inference guided by dual physical and spectral constraints. Evaluations on three standard benchmarks (nine-species, seven-species, and HC-PT) demonstrate that PhysNovo achieves state-of-the-art performance. The method significantly outperforms existing baselines in both amino acid and peptide-level precision and exhibits excellent generalization to unseen species, as validated through leave-one-out cross-validation.

**Compliance With Llm Reviewing Policy:**

Affirmed.

**Final Justification:**

The rebuttal has addressed my main concerns.

**Key Questions For Authors:**

1.Could you provide the standalone accuracy of the precursor-guided MLP for length prediction? Furthermore, how sensitive is the model's final peptide-level precision to errors in the initial length estimate? Does the discrete diffusion framework possess an inherent ability to recover from $\pm 1$ or $\pm 2$ length prediction errors? Since the process involves $T=100$ denoising steps, an inaccurate initial length could act as a strict bottleneck. Providing these metrics would clarify whether this dependency is a critical vulnerability or a challenge that has been effectively addressed. Strong performance in this area would significantly increase my confidence in the reliability of the method.

2.Have you considered integrating similar physical constraints into the comparative baselines? While PhysNovo’s performance gains are impressive, it is currently difficult to distinguish the advantages of the discrete diffusion paradigm itself from the benefits of simply enforcing strict physical constraints. If the authors can demonstrate that this constraint is uniquely effective within your framework, it would strongly validate the paper’s claims regarding architectural innovation.

3.How does the computational complexity of the knapsack-based feasibility kernel scale as the vocabulary size expands? Current experiments utilize 20 standard amino acids and 3 common PTMs. In an Open Search scenario involving hundreds of potential modifications, can the bounded reachability approximation still prune the search space efficiently and accurately without introducing significant latency?

4.The paper strongly argues against the post-hoc filtering used in recent NAR models, such as $\pi$-PrimeNovo. Could you provide a direct quantitative comparison of inference latency (e.g., spectra processed per second) between PhysNovo and these state-of-the-art NAR + post-processing methods? Given that industrial proteomics prioritizes high throughput, demonstrating that PhysNovo’s 100-step iterative optimization is computationally competitive with post-hoc NAR approaches would directly address concerns regarding its practical utility.

**Limitations:**

See Questions.

**Strengths And Weaknesses:**

Strengths
1. The paper is technically robust and proposes a physics-grounded approach to a well-defined problem in proteomics. The underlying premise of modeling peptide sequencing as an iterative refinement process via discrete diffusion, rather than a simple translation task, is well-justified. The authors correctly identify that auto-regressive (AR) models suffer from error propagation and directional bias, while non-auto-regressive (NAR) models, such as PrimeNovo, struggle to maintain physical mass validity. The proposed "PhysNovo" effectively addresses these limitations by embedding hard mass constraints directly into diffusion transitions using a knapsack-based feasibility kernel.
2. The authors evaluate their method on three standard benchmarks (nine-species, seven-species, and HC-PT), which represent a variety of spectral resolutions and biological contexts. The comparison is extensive, covering a wide range of baselines, including current SOTA methods. Furthermore, the ablation studies strongly confirm the necessity of the proposed components; specifically, removing hard mass constraints leads to a significant drop in peptide precision, validating the hypothesis that a mass-conserving framework effectively narrows the solution space.
3. The paper is clearly written and logically structured. The related work section provides a thorough review of the evolution of de novo sequencing and constrained diffusion models, clearly differentiating the current work from existing literature and articulating its specific innovations.
4. The paper  demonstrates high originality by combining discrete diffusion with combinatorial optimization, introducing a novel methodology to the field of bioinformatics.

Weaknesses
1. The entire generation process is heavily reliant on the initial peptide length predicted by the MLP. If the length prediction is incorrect, subsequent mass constraints and iterative refinements cannot correct this error. However, the paper neither reports the accuracy of the length prediction module nor evaluates how length errors affect final sequencing performance. The reliability of this core prerequisite remains unverified.
2. The primary performance gains may stem from the embedded mass constraints, which appear to be a transferable module. Since the authors only applied these constraints to PhysNovo and not to the SOTA baseline methods, it is difficult to determine whether the improvements arise from the diffusion paradigm itself or simply from the mass constraint module.
3. While embedding mass constraints into diffusion steps is innovative, the fundamental idea of using precursor ion mass to filter invalid sequences is a well-established technique that has been widely used in previous de novo sequencing methods.
4. PhysNovo implements physical constraints by calculating the remaining mass budget for each candidate amino acid. In the current experimental setup, the vocabulary is limited to 20 standard amino acids and three common PTMs. However, real biological samples can contain hundreds of potential variations. If the method is extended to Open Search or scenarios with numerous complex PTMs, the search space for validating all possible combinations during each denoising iteration could grow exponentially. It is questionable whether the current bounded reachability mechanism can remain efficient and accurate in such high-complexity scenarios.

---

> ### Author Rebuttal · Authors · 2026-03-30
>
> We thank the reviewer for the insightful feedback. We address your concerns below, and will include all these discussions in the final revision.
>
> **Length Prediction and Sensitivity Analysis. (W1, Q1)**
> We have provided an analysis and the standalone accuracy table in our response to **Reviewer jiuq--(W1, W2, Q1)**. Briefly, the predictor achieves 94.2% exact match, 98.4% within ±1, and 99.1% within ±2 residues. To assess sensitivity, we analyze performance under length error $\Delta L = |\hat{L} - L^*|$:
> |Length Error ($\Delta L$)|Fraction|AA-level Prec.|Peptide-level Prec.|
> |:---:|:---:| :---:|:---:|
> |0 (Exact match)|94.2%|0.839|0.624|
> |1|4.2%|0.531|0.000|
> |$\ge$ 2|1.6%|0.149|0.000|
>
> As expected, incorrect length leads to zero peptide-level precision due to the requirement of exact sequence matching. Because discrete diffusion operates in a fixed-dimensional space, the model cannot correct length after initialization and instead optimizes within the predicted length using MS1 and MS2 constraints. Importantly, degradation is graceful rather than catastrophic. For $\Delta L = 1$, because $\pm 1$ errors often arise from local isobaric substitutions (e.g., `N` vs. `GG`), the diffusion process still actively aligns the rest of the sequence with the MS2 spectral evidence, correctly recovering a portion of the sequence (**0.531 AA prec.**). Even for $\Delta L \ge 2$, terminal fragment alignment yields non-trivial precision (0.149), indicating robustness of MS2 conditioning. Finally, we argue that length prediction is not a strict bottleneck: prediction errors affect <6% of samples, and peptide-level precision with correct length ($\Delta L = 0$) is 0.624. Thus, even a perfect predictor would only improve overall precision from 0.588 to 0.624 (+0.036), demonstrating the adequacy of the current module.
>
> **Diffusion vs. Constraints. (W2, Q2)**
> Existing baselines *already* enforce strict MS1 hard mass constraints, but necessarily as **post-hoc filters** (e.g., CasaNovo, $\pi$-PrimeNovo), since left-to-right or one-shot paradigms cannot incorporate a dynamic global constraint during generation. We note that completely ablating all spectral inputs to test a "pure" backbone is infeasible: unlike AR models, discrete diffusion starts from fully masked tokens. Removing all conditioning would degenerate the model into an unconditional random peptide generator.
> To disentangle generative paradigm vs. constraints, we refer to our ablations (**Table 5** in main paper): **Advantage of the Diffusion Paradigm (ID 2):** Relying purely on soft MS2 spectral conditioning, this "pure diffusion" variant still yields a peptide precision of **0.577**. This vastly outperforms the CasaNovo (**0.481**), which *already* incorporates a post-hoc MS1 mass filter. This proves that the bidirectional diffusion paradigm is fundamentally superior on its own. **Advantage of In-Generation Constraints (ID 1):** Conversely, using only the MS1 constraint, **PhysNovo** achieves **0.583**, again outperforming CasaNovo (**0.481**). This confirms that integrating physical constraints during generation is more effective than post-hoc filtering.
> Together, these results show diffusion is fundamentally stronger, while the Knapsack kernel provides the final synergy to achieve SoTA (0.588).
> Finally, peptide sequencing is inherently adversarial to AR generation. When the mass spectra exhibit weak N-terminal signals but strong C-terminal signals, an AR model must guess the ambiguous N-terminus first. An early error irrevocably consumes the wrong mass budget, forcing incorrect predictions at the C-terminus despite clear evidence there, as AR cannot retroactively revise past tokens. In contrast, diffusion refines sequences jointly, first locking in confident regions (e.g., C-terminus), then resolving ambiguous parts under global mass constraints—highlighting the unique advantage of iterative, bidirectional refinement.
>
> **Novelty. (W3)**
> While both constraint mechanisms and diffusion paradigms have been explored in prior work, their joint application to *de novo* peptide sequencing is fundamentally challenging. Historically, mapping MS/MS spectra to peptide sequences has naturally aligned with the sequence-to-sequence (AR) translation paradigm, whereas diffusion requires generating discrete peptide sequences from noise conditioned on complex, continuous spectra—a significant cross-modal challenge. Our key contribution is bridging this gap: through spectral cross-attention and the MS1 knapsack kernel, PhysNovo converts physical evidence into a guided denoising process. This provides a principled alternative that mitigates error accumulation in AR models and reduces reliance on heuristic post-processing in AR and NAR methods.
>
> **Scalability to Open Search, Complex PTMs, and Isomers. (W4, Q3)**
> See our discussion for **Reviewer jiuq--(W4, Q4)**.
>
> **Inference Latency. (Q4)**
> See our discussion for **Reviewer jiuq--(Q3)**.

---

> > ### Author Rebuttal · Reviewer_6dbg · 2026-04-03
> >
> > Thank you for your detailed responses. They have addressed my remaining concerns, and I do not have further questions. I have therefore raised my score.

---

> > > ### Author Response · Authors · 2026-04-06
> > >
> > > We sincerely appreciate the reviewer’s constructive feedback and thoughtful suggestions. We are glad that our responses have adequately addressed the concerns raised, and we are grateful for the reviewer’s support in improving our work.

---

### Official Review · Reviewer_2yH5 · 2026-03-08

**Soundness:** 3
**Presentation:** 3
**Significance:** 2
**Originality:** 2
**Overall Recommendation:** 3
**Confidence:** 4

**Summary:**

The paper applies discrete diffusion to de novo peptide sequencing from tandem mass spectrometry data. Unlike other approaches, the discrete diffusion models are non-autoregressive and are physically constrained to respect mass conservation in the generation process. The key methodological contribution is in using a knapsack-based feasibility constraint in the generation process.  The paper claims that these approaches enable better generalization and accuracy compared to autoregressive and non-autoregressive approaches.

**Compliance With Llm Reviewing Policy:**

Affirmed.

**Key Questions For Authors:**

1) It is somewhat unclear how much better the paper’s approach is over the baselines. How significant is the improvement in precision/recall? Can you add error bars and standard deviation to the Tables?

2) The inductive bias of AR models over non-autoregressive models seems intuitive at face value, but the claim that non-autoregressive models are superior is only substantiated with higher precision/recall of peptide sequencing. It is a bit unclear what aspect non-autoregressive generation helps with here. Is it the bias of the prior, the process of iterative refinement itself, or both? Are there peptide sequences that are inherently adversarial to a left-to-right generation order?

3) The claim that physical validity is preserved is substantiated with an ablation of the mass constraint, which results in reduced precision/recall, but this seems to be only a marginal reduction. How often and to what degree does the generative model violate the mass constraint otherwise?

4) While the paper uses hard constraints, other approaches apply guidance techniques to condition on physical properties [1,2]. Are there any inherent advantages of a hard physical constraint over guidance-like conditioning?

5) In equation (9), the way the loss is defined can bias the algorithm to shorter sequences since the sequence length should also be jointly estimated. Would that be of any concern in practice? Would it make sense to estimate L independently? Would this be a limitation that AR models do not suffer from?

**Limitations:**

The main limitation, as discussed in my notes above, is the significance of the results as compared to the existing techniques.

**Strengths And Weaknesses:**

Strengths: The paper addresses an important problem of de novo mapping of mass spectrometry (MS) data to protein sequences. This is an active and somewhat crowded area with several existing computational methods, as the authors outline. The main novelty of the work lies in the use of diffusion models, which avoid the directional constraints of autoregressive (AR) models when generating sequences. A strength of the paper is the integration of several well-motivated components, such as an MS peak encoder, a knapsack kernel, an ESM protein language model, and discrete diffusion, within a single unified framework. Each component is justified and appropriate for the task.

Weaknesses: The main weakness is that the proposed method does not appear to significantly outperform existing approaches, despite the claims made in the introduction and contributions. Results in Tables 1–3 show only marginal improvements over prior methods. The absence of error bars or standard deviations further makes it difficult to assess the statistical significance of the reported gains. As a result, it remains unclear whether the observed improvements are truly due to the new modeling approach. Another limitation is that, while the method combines several techniques in a novel way, most of the individual components already exist in prior work. Overall, the contribution is clearly novel from a systems or integration perspective, but somewhat less so from a fundamental machine learning or algorithmic standpoint.

Soundness: The motivation and method are sound. The individual components are well motivated.

Presentation: The paper presentation is clear to follow and does not suffer from any obvious weaknesses.

Significance: The paper addresses a problem in de novo peptide sequencing from tandem mass spectrometry data, which seems to be a significant problem of interest to solve with deep learning techniques. However, it seems unclear whether the proposed solution improves over baseline approaches significantly. For this reason, the advantages of reducing bias over AR models are also unclear + AR models would not need to know the sequence length a priori.

Originality: The paper applies discrete diffusion models to generate a prior for reconstructing the peptide sequence from mass spectrometry measurements. The paper also introduces additional methods of adding physical constraints to discrete diffusion models and a re-masking strategy based on confidence to correct for errors accumulated in the trajectory. A novel contribution appears to be the addition of strict physical constraints to discrete diffusion generation. The re-masking sampler is somewhat similar to previous works [1,2], which prescribe confidence-based sampling and re-masking procedures.

[1] https://arxiv.org/pdf/2502.06768
[2] https://arxiv.org/pdf/2407.21243

---

> ### Author Rebuttal · Authors · 2026-03-30
>
> We thank the reviewer for the insightful feedback. We address your concerns below, and will include all these discussions in the final revision.
>
> **Improvements and Variance. (W1, Q1)**
> |Dataset|Method|AA Prec.|AA Recall|Peptide Prec.|Peptide AUC|
> |:-|:-:|:-|:-|:-|:-|
> |Nine-species|PhysNovo|0.815 $\pm$ 0.002|0.814 $\pm$ 0.002 |0.588 $\pm$ 0.003|0.553 $\pm$ 0.003|
> |Seven-species|~|0.568 $\pm$ 0.003|0.569 $\pm$ 0.003|0.346 $\pm$ 0.004|0.290 $\pm$ 0.003|
> |HC-PT|~|0.649 $\pm$ 0.002|0.652 $\pm$ 0.003|0.473 $\pm$ 0.003|0.444 $\pm$ 0.004|
>
> Across 5 runs, PhysNovo shows high stability with deviations of $\pm 0.002$–$0.004$. The observed gains (e.g., +1.8% AA Prec. on Nine-species) are an order of magnitude larger, confirming statistical significance. Although some improvements appear small, de novo sequencing requires exact full-sequence matches, even a 1-2% absolute gain in full peptide accuracy represents a substantial improvement in recovering usable proteins. Notably, PhysNovo achieves substantial improvements in PTM identification, surpassing SoTA by +5.0% prec. and +5.1% recall on Seven-species, and +3.1% prec. on HC-PT. This reflects enhanced sensitivity to subtle mass shifts and stronger performance on complex PTMs.
>
> **Diffusion vs. Constraints. (W2, Q2)**
> See our discussion for **Reviewer 6dbg--(W2, Q2)**.
>
> **Novelty. (W3)**
> See our discussion for **Reviewer 6dbg--(W3)**.
>
> **Impact of Mass Constraints. (Q3)**
> The marginal difference in precision does not reflect the critical impact of the knapsack constraint on physical validity. While the soft-conditioned model (ID 2) can recover correct sequences in some cases, the primary role of the knapsack constraint, however, is to regulate incorrect predictions. On the Nine-species dataset:
> |Model Configuration|Peptide Prec.|Mass Violation Rate| Degree of Violation (Da)|
> |:-|:-:|:-:|:-:|
> |PhysNovo (Full Model)|0.588|0.0%|0.0|
> |PhysNovo (ID 2, Soft-only)|0.577|18.4% (In Incorrect Predictions)|$\approx$ 117.1 ($\approx$ 1 residue)|
>
> Without the MS1, 18.4% of incorrect sequences violate mass conservation, with an average deviation of ~117.1 Da, indicating hallucinated residue compositions. The full model enforces mass consistency, ensuring all outputs satisfy MS1 constraints. Even if it fails to recover the completely accurate sequence, the output remains a physically valid candidate solution. Thus, beyond modest precision gains, the MS1 is essential for guaranteeing physically valid and reliable predictions.
>
> **Hard Constraint vs. Guidance. (Q4)**
> PhysNovo offers two key advantages:
> 1) Absolute Guarantee & *A Priori* Pruning: Guidance methods[1,2] reshape probabilities but cannot ensure strict compliance with MS1 mass, an inviolable physical constraint. They may produce physically infeasible sequences requiring post-hoc filtering. In contrast, our hard constraint enforces exact mass matching by pruning invalid branches at each step, avoiding wasted computation.
> 2) Synergy with Guidance-like Correctors: PhysNovo actually synergizes hard constraints with guidance-like mechanisms. We enforce the absolute MS1 mass via hard constraints, and employ a **guidance-like informed corrector** (iterative remasking) to align with the MS2 evidence. This revises uncertain tokens, sharing the core philosophy of [1, 2].
>
> **Bias Toward Shorter Sequences. (Q5)**
> The proposed loss does not bias toward shorter sequences. Sequence generation and length prediction are jointly trained but decoupled: the diffusion loss (Eq.9) operates on tokens conditioned on a fixed length, while length is supervised separately. During training, ground-truth length is known and fixed, and loss is computed only on non-padded positions, so optimization occurs within each sequence without cross-length comparison, thereby avoiding any systematic preference toward shorter outputs.
> Empirical results confirm this conclusion: predicted lengths closely match ground truth (13.4 vs. 13.5), with 94.2% exact match and minimal deviation (2.7% shorter, 3.1% longer).
> |Metric|Value|
> |:-|:-|
> |Avg Length (Ground Truth)|13.4|
> |Avg Length (Predicted $\hat{L}$ \& Generated)|13.5|
> |Shorter than Ground Truth|2.7%|
> |Longer than Ground Truth|3.1%|
> |Exact Match|94.2%|
>
> Independent length estimation is necessary. Unlike AR, discrete diffusion requires a fixed state space throughout the denoising process, making it necessary to determine the sequence length prior to generation. Accordingly, at inference, we use a highly accurate length predictor $P_\phi(L \mid \mathbf{c})$ (94.2% exact match), ensuring that generation is performed under a reliable length hypothesis. Unlike AR models relying on [EOS] termination, PhysNovo separates length estimation from sequence generation, avoiding premature or delayed stopping while maintaining superior performance over AR baselines (Tab. 1 in main paper). Combined with the high accuracy of the length predictor, demonstrating that independent length modeling is a reliable and effective approach.

---

> > ### Author Rebuttal · Reviewer_2yH5 · 2026-04-01
> >
> > Thanks for the notes. I don't have further questions. The score mostly reflects the limited performance gain and the methodological novelties.

---

> > > ### Author Response · Authors · 2026-04-01
> > >
> > > We sincerely appreciate your feedback and suggestions. We are glad to learn that our rebuttal has addressed your concerns. Please do not hesitate to reach out if further clarification is needed. Should you find our responses satisfactory, we would be grateful if you would consider updating your score accordingly to support this work.

---

### Official Review · Reviewer_jiuq · 2026-03-12

**Soundness:** 3
**Presentation:** 3
**Significance:** 3
**Originality:** 3
**Overall Recommendation:** 4
**Confidence:** 4

**Summary:**

This paper tackles the task of de novo peptide sequencing from tandem mass spectrometry (MS/MS) data by proposing a discrete diffusion generative framework named PhysNovo. Existing autoregressive (AR) and non-autoregressive (NAR) models often struggle to fully satisfy MS1 mass conservation constraints during generation, frequently yielding chemically invalid sequences. To address this, the authors reformulate peptide sequencing as a discrete diffusion inference process strictly bounded by physical mass constraints.

**Compliance With Llm Reviewing Policy:**

Affirmed.

**Final Justification:**

The authors’ rebuttal has addressed my concerns; therefore, I will maintain my current positive score.

**Key Questions For Authors:**

1. Regarding the model's assumptions on fixed dimensionality and exact MS1 mass constraints: (1) Can the model accurately predict the true sequences for peptides with unknown lengths or those containing complex modifications and cross-links? (2) In the context of real-world samples featuring chimeric spectra, is relying on a single MS1 precursor mass constraint still sufficient? Are there any planned optimizations or directions to address this specific scenario?

2. During the training and fine-tuning phases, the masking follows a random uniform distribution, whereas a fixed threshold of $$0.7$$ is applied during inference. Although the appendix claims this does not cause a severe distribution shift, given the critical impact of the threshold on model performance, could the authors provide additional ablation studies? This would more solidly justify the rationale and stability of using this fixed threshold during inference.

3. Beyond the existing theoretical comparisons of computational complexity and time steps, could the authors provide a more concrete benchmark based on actual wall-clock time? Specifically, it would be highly valuable to demonstrate the practical inference efficiency of the proposed model compared to baselines when processing large-scale, high-throughput proteomics data.

4. Considering the previously mentioned limitations, scaling the model to incorporate a massive number of complex PTMs in the future will subject the Knapsack-based kernel to a severe curse of dimensionality and interference from isomers. Do the authors currently have any anticipated optimization plans or algorithmic strategies to mitigate this exponential explosion of the combinatorial search space?

**Limitations:**

yes

**Strengths And Weaknesses:**

Strength:

1. This paper reformulates the complex \textit{de novo} sequencing pipeline into a discrete diffusion process under physical constraints. This avoids the sequential fragment generation of autoregressive models, making it algorithmically more aligned with the physical reality of bidirectional fragmentation in mass spectrometry, and also eliminates the early error accumulation commonly seen in sequence models.

2. By incorporating a low-complexity Knapsack-based kernel, it achieves hard constraints on the fragments. This ensures that the total mass of the generated candidate peptides strictly complies with actual physical mass conservation, avoiding the hallucination of generating chemically invalid sequences.

3. Utilizing global spectral conditions and an adaptive low-confidence re-masking strategy, the model gains a "self-correction" capability. This greatly improves the model's generalization confidence on low signal-to-noise ratio (SNR) and unseen species data, while maintaining a lower parameter count.

Weakness:

1. Discrete diffusion models fix the dimensionality during the denoising process, and therefore may heavily rely on an accurate prior prediction of the peptide length.

2. The Knapsack-based kernel hard constraints adopted by the model rely on the ideal assumption of an accurate and unique MS1 mass. For peptides with unknown lengths or complex modifications and cross-links, it remains uncertain whether the model can reliably predict the true sequence that matches the exact spectral pattern.

3. When processing chimeric spectra in real-world samples (i.e., where a single MS/MS spectrum contains fragments from multiple MS1 precursor ions) or dealing with peptides of unknown lengths, complex modifications, and cross-links, this singular constraint might limit the model's applicability in complex real-world scenarios.

4. The current model has only been validated on a 23-token vocabulary containing 3 common post-translational modifications (PTMs). However, real biological environments feature a wide variety of complex modifications. If one attempts to scale up the model's vocabulary to include a massive number of complex PTMs or non-natural amino acids, the combinatorial search space of the Knapsack-based kernel would expand exponentially. Additionally, the widespread presence of isomers with extremely subtle mass differences would significantly dilute and weaken the effectiveness of the current hard constraint kernel. This is a non-negligible bottleneck for scaling the model to broader applications.

---

> ### Author Rebuttal · Authors · 2026-03-30
>
> We thank the reviewer for the insightful feedback. We address your concerns below, and will include all these discussions in the final revision.
>
> **Predicting Peptide with Unknown Lengths, PTMs, and Cross-Links. (W1, W2, Q1)**
> **Length Prediction:** As shown below, the standalone length predictor is highly accurate (94.2% exact match, 98.4% within ±1, 99.1% within ±2), indicating $P_\phi(L \mid c)$ is sharply concentrated around the true length. Furthermore, as detailed in sensitivity analysis (Reviewer 6dbg), this fixed-length dependency is not a practical bottleneck: rare ±1 errors lead to graceful degradation rather than failure, with the model still recovering substantial sequence information (0.531 AA prec.) via MS2 conditioning.
> |Metric|Definition|Accuracy|
> |:-|:-|:-:|
> |Exact Match|$\hat{L} = L^*$|94.2%|
> |±1|$L^* \in[\hat{L}-1, \hat{L}+1]$|98.4%|
> |±2|$L^* \in [\hat{L}-2, \hat{L}+2]$|99.1%|
>
> **Complex PTMs:** The Knapsack kernel does not assume absolute mass equality; it operates within standard MS1 mass tolerances (ppm-level $\epsilon$), aligning with conventional *de novo* settings. Within this tolerance, the constraint is actually the key to handling PTMs, rather than a limitation. By enforcing the global mass budget at each diffusion step, PhysNovo avoids cumulative mass drift seen in AR models and effectively captures PTMs, as reflected in improved PTM-level precision (e.g., +5.0% on Seven-species).
> **Cross-Links:** Cross-linked peptides indeed break the single-sequence mass conservation assumption. This remains a fundamental challenge for the vast majority of contemporary *de novo* sequencing models (e.g., LIPNovo, and $\pi$-PrimeNovo), which are inherently designed for linear peptides. Like these SoTA baselines, PhysNovo currently focuses on linear peptides. Extending the state space to model inter-linked topologies is an important, orthogonal research direction for PhysNovo and the entire field.
>
> **Chimeric Spectra and Optimizations. (W3, Q1)**
> Applying a single MS1 constraint to chimeric spectra is a limitation shared across existing methods. However, our conditional formulation naturally provides a principled extension: we extract multiple candidate masses ($\{\\{c\_i\\}\}\_{i=1}^K$) from the MS1 isolation window and evaluate $p_\theta(\mathbf{y}_i \mid \mathbf{x}, \mathbf{c}_i)$ for each. This yields multiple hypotheses under distinct mass constraints. The Knapsack constraint restricts each solution to sequences matching $\mathbf{c}_i$, thereby filtering out inconsistent fragment ions and effectively reducing interference from co-eluting peptides.
>
> **Threshold Ablation. (Q2)**
> We refer the reviewer to Sec. 4.4 (Table 6), where we have provided an ablation on $\tau$. Performance is stable for $\tau \in [0.6, 0.8]$ and peaks at 0.7, while a higher value (0.9) degrades results. This empirically supports $\tau=0.7$ as a stable and optimal choice.
>
> **Inference Efficiency. (Q3)**
> Conducted on a single RTX 4090 GPU. PhysNovo outperforms the AR baseline (Casanovo) in both speed and accuracy (0.815 vs. 0.697). Although iterative refinement is slower than the NAR method ($\pi$-PrimeNovo), it achieves higher accuracy (0.815 vs. 0.790). Critically, reducing to $T=50$ yields near-NAR throughput (99 vs. 105) while retaining superior accuracy (0.801 vs. 0.790), demonstrating a favorable speed–accuracy trade-off and practical efficiency for high-throughput applications.
> |Model|Speed (spectra/s)|AA Prec.|
> |:-|:-:|:-:|
> |Casanovo V2 (AR, Post-hoc)|11|0.697|
> |$\pi$-PrimeNovo (NAR, Post-hoc)|105|0.790|
> |PhysNovo ($T=100$)|46|**0.815**|
> |PhysNovo ($T=50$)|**99**|0.801|
>
> **Scalability to Open Search, Complex PTM, and Isomers. (W4, Q4)**
> The knapsack-based feasibility kernel does not exhibit exponential complexity in practice. Its bounded reachability reduces to a continuous $\mathcal{O}(1)$ scalar check per token (Eq. 11), For a peptide of length $L$ over $T$ denoising steps, the overall worst-case complexity is strictly bounded by $\mathcal{O}(T \cdot L \cdot |\mathcal{V}|)$, as it depends only on global mass bounds ($w_{\min}, w_{\max}$). Thus, expanding the PTM vocabulary incurs only linear scaling. We agree, however, that extreme large-scale PTM settings introduce additional challenges. We propose compatible strategies: (1) mass-grouped compression to reduce effective vocabulary size, (2) cache reuse across denoising steps, (3) extending pretrain to a broader PTM pretrain further concentrates probability mass on plausible candidates, reducing the search space. For isomers, the MS1 constraint acts as a feasibility filter, while final discrimination relies on MS2-conditioned likelihood, this complementary design allows the model to resolve ambiguities organically. Although indistinguishable isomers (e.g., Leu/Ile) remain a fundamental limitation of mass spectrometry, our framework maximizes spectral utilization. Overall, the kernel scales linearly and supports practical optimization under complex PTM settings.

---

> > ### Author Rebuttal · Reviewer_jiuq · 2026-04-03
> >
> > My issue has already been resolved. Given that the current score is positive, I will keep the original score.

---

> > > ### Author Response · Authors · 2026-04-06
> > >
> > > We sincerely thank the reviewer for their constructive feedback and encouragement. We are delighted that our responses have fully addressed your concerns, and we truly appreciate your support in helping us improve our work.

---

### Decision · Program_Chairs · 2026-04-30

**Decision:**

Accept (regular)

**Comment:**

This paper presents PhysNovo, a new generative framework that reformulates de novo peptide sequencing as an iterative discrete diffusion process that respect physical mass constraints and overcome directional bias in previous autoregressive methods. While the majority of the reviewers note some limitations in ML novelty (e.g., individual components exist in prior work) and relatively small performance gain compared to baselines, they believe the application domain is novel and important, and the integration of Knapsack-based kernel is nontrivial. The reviewers raised important questions regarding the reliance on a strong length predictor, and the authors successfully addressed that with evidence that performance degradation is graceful under length error. Furthermore,  they provided evidence that in the exponential search space of de novo sequencing, even modest 1-2% absolute gains represent substantial improvements in protein recovery, especially for post-translational modifications.

Overall, while the mL components are known, their systems-level integration into a strictly physics-grounded alternative for proteomics is original and effective, and demonstrates strong potential of diffusion method in this area. Therefore, the merits of this work outweigh the remaining weaknesses, making it a valuable contribution to the ICML community.